# PROVABLE POST-DEPLOYMENT DETERIORATION MONITORING

## ABSTRACT

Data distribution often changes when deploying a machine learning model into a new environment, but not all shifts degrade model performance, making interventions like retraining unnecessary. This paper addresses model post-deployment deterioration (PDD) monitoring in the context of unlabeled deployment distributions. We formalize unsupervised PDD monitoring within the model disagreement framework where deterioration is detected if an auxiliary model, performing well on training data, shows significant prediction disagreement with the deployed model on test data. We propose D-PDDM, a principled monitoring algorithm achieving low false positive rates under non-deteriorating shifts and provide sample complexity bounds for high true positive rates under deteriorating shifts. Empirical results on both standard benchmark and a real-world large-scale healthcare dataset demonstrate the effectiveness of the framework in addition to its viability as an alert mechanism for existing high-stakes ML pipelines.

## 1 INTRODUCTION

Performance guarantees of conventional machine learning (ML) models hinge on the belief that the distribution of data with which these models train is identical to the distribution on which they are deployed (Rabanser et al., 2019; Recht et al., 2018; Santurkar et al., 2020). In many real-world scenarios such as healthcare, however, this assumption fails due to distribution shift during model deployment. Benchmarks such as WILDS (Koh et al., 2021) and WILD-Time (Yao et al., 2022) have encouraged machine learning researchers to study and better understand how data shifts influence predictive systems. Yet, the number of tools at a practitioner's disposal for the creation of predictive models far exceed those to monitor model failures. There is a need to create *guardrails* that *self-detect* and *alert* end-users to critical changes in the model when its performance drops below acceptable thresholds (Habib et al., 2021; Zadorozhny et al., 2022).

Post-deployment deterioration (PDD) monitoring presents a distinct set of systemic challenges stemming from considerations over the feasibility of deployment in real-world ML pipelines. Predominant is the scarcity of labels during deployment: for many downstream tasks such as in healthcare, labels are expensive to obtain (Razavian et al., 2015) or require human intervention (Liu et al., 2022). Moreover, due to deployed models predicting events temporally extended in the future (Boodhun & Jayabalan, 2018; Zhang et al., 2019), labels might even be unavailable. Another systemic challenge is the robustness of the monitoring system, that the latter should be flagging critical changes in deployment efficiently as early as a few samples, and that it should remain robust to non-deteriorating changes as to minimize unnecessary interruptions of service among other practical considerations.

To address these challenges, we conceive a set of desiderata for any algorithm monitoring PDD, targeting their practicality and effectiveness as plug-ins to ML pipelines. To address the scarcity of labels, PDD monitoring algorithms should operate on unlabeled data from the test distribution to ascertain potential deterioration of the deployed model. PDD monitoring algorithms should not depend on training data during deployment. Continuous (even indefinite) access to sensitive or personally identifiable training data might violate certain regulations protecting the privacy of data subjects (Mühlhoff, 2023). An algorithm satisfying this desideratum is thus a scalable algorithm as it functionally only audits its input stream during monitoring with minimum data storage and regulatory concerns. Finally, PDD monitoring algorithms should be robust to flagging non-deteriorating changes and effective in few-shot settings.

Insofar as designing monitoring protocols satisfying the above, recent related works only partially attend to individual desiderata. The literature on distribution shifts while achieving strong empirical performance on unlabeled deployment data (Liu et al., 2020; Zhao et al., 2022), are not robust to false positives when the distribution shift is non-deteriorating. The model disagreement framework (Chuang et al., 2020; Jiang et al., 2021; Ginsberg et al., 2023; Rosenfeld & Garg, 2023) emerges as the natural setup for monitoring with downstream performance considerations via the tracking of disagreement statistics, while foregoing explicit distribution shift computations. However, shift-based and disagreement-based monitoring methods all depend on the presence of training data post-deployment, and do not provide any guarantees on robustness against false positives in the monitoring of non-deteriorating shifts.

In this paper, we answer all desiderata for PDD monitoring via the disagreement framework by proposing **D**isagreement-based **P**ost-**D**eployment **D**eterioration **M**onitoring (**D-PDDM**), a novel algorithm operating in the unsupervised deployment setting (1), requiring no training data during monitoring (2), and is provably robust in flagging deteriorating shifts as well as resilient to flagging non-deteriorating shifts (3). A comparison of the satisfaction of PDD desiderata of our method with related work in the literature is provided in Tab. 1. Our contributions are as follows:

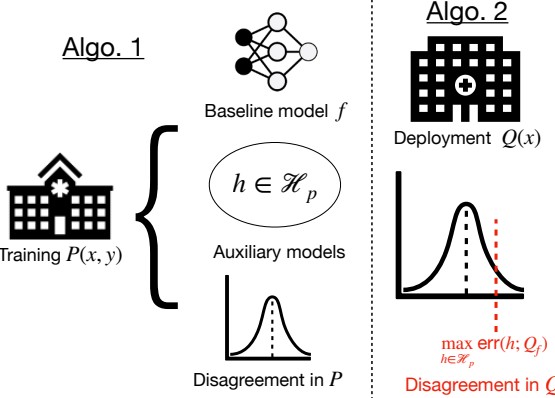

**Figure 1:** Our protocol consists of two steps – pre-training and **D-PDD M**onitoring. Importantly, monitoring does not require training data $P(x)$ for unsupervised deployment data $Q(x)$.

**Disagreement based PDD (D-PDD).** We formulate the unsupervised PDD problem as the model disagreement framework (D-PDD), requiring no training data during deployment and is model-agnostic. We further demonstrate the conditions for which model disagreement is equivalent to PDD.

**Provable algorithm.** We propose **D-PDDM**, a theoretically principled algorithm to monitor D-PDD. We prove that in the presence of deteriorating shift, D-PDDM provably monitors D-PDD in finite samples, and in the presence of non-deteriorating shifts, D-PDDM achieves low false positive rates. In addition, we broadly characterize various regimes of shift where D-PDDM might not be effective, and suggest alternative methods to circumvent these limitations.

**Empirical validation.** Our experimental results on various shift scenarios in both benchmark and real-world large-scale healthcare – General Medicine INpatient Initiative (GEMINI) dataset (Verma et al., 2021). When comparing the proposed method with other popular monitoring baselines, our method effectively detects the D-PDD with low false positive rates (FPR) when shifts are non deteriorating, and achieves better true positive rates (TPR) when shifts are deteriorating. Further, owing to the decoupling of the algorithmic protocol into two stages, D-PDDM is *efficiently scalable in the size of the training dataset*, a critical consideration on the feasibility of its application onto current ML pipelines that is not enjoyed by standard baselines.

**Table 1:** Comparisons between related work. *Training data-free*: whether post-deployment monitoring requires training data; *Deteriorating*: whether the method provably monitors the deteriorating shift; *Non-deteriorating*: whether the method is provably robust in the non-deteriorating shift; *Disagreement*: whether the method is based on the disagreement framework.

| | Training data-free | Deteriorating | Non-deteriorating | Disagreement |
|---|---|---|---|---|
| Liu et al. (2020) | ✗ | ✗ | ✗ | ✗ |
| Jiang et al. (2021) | ✗ | ✗ | ✗ | ✓ |
| Zhao et al. (2022) | ✗ | ✗ | ✗ | ✗ |
| Rosenfeld & Garg (2023) | ✗ | ✓ | ✗ | ✓ |
| Ginsberg et al. (2023) | ✗ | ✓ | ✗ | ✓ |
| **D-PDDM (ours)** | ✓ | ✓ | ✓ | ✓ |

## 2 PROBLEM SETUP

Consider a function class $\mathcal{H}$ of $h : \mathcal{X} \rightarrow \mathcal{Y} = \{0, 1\}$. We use $g \in \mathcal{H}$ to denote the ground truth labeling function, $f$ to denote the deployed classifier, and $h$ as the auxiliary classifier. We denote the marginal distribution w.r.t $x$ as $\boldsymbol{P}_x(x)$ and the joint distribution with the labeling function in the subscript. For a data distribution $\boldsymbol{P}_x$ over the domain $\mathcal{X}$ and any labeling function $f(x)$, we define the joint distribution as $\boldsymbol{P}_f = \boldsymbol{P}_f(x, y) = \boldsymbol{P}_x(x, f(x))$. Let the population error and its corresponding empirical counterpart with $n$ samples $\mathcal{D}^n = \{(x_i, y_i)\}_{i=1}^n$ be

$$\text{err}(f; \boldsymbol{P}_g) \coloneqq \Pr_{x,y \sim \boldsymbol{P}_g} [f(x) \neq y], \qquad \widehat{\text{err}}(f; \mathcal{D}^n) \coloneqq \widehat{\text{err}}(f; \boldsymbol{P}_g) \coloneqq \frac{1}{n} \sum_{i=1}^n |f(x_i) - y_i|$$

**Training and deployment distribution.** We denote $\boldsymbol{P}_x$ as the training (marginal) distribution, and $\boldsymbol{Q}_x$ as the deployment distribution. We assume data is i.i.d. sampled from both $\boldsymbol{P}_x$ and $\boldsymbol{Q}_x$. We consider that $n$ labeled samples from $\boldsymbol{P}_g$ are available before deployment, and $m$ unlabeled samples from $\boldsymbol{Q}_x$ are obtained during the deployment.

**Disagreement.** For any two functions $f$ and $h$ in $\mathcal{H}$, we say that they disagree on any point $x \in \mathcal{X}$ if $f(x) \neq h(x)$. Given the binary classification setting, we can write the disagreement rate of the function $h$ with $f$ on distribution $\boldsymbol{Q}_x$ in terms of error as $\text{err}(h; \boldsymbol{Q}_f)$ or $\text{err}(f; \boldsymbol{Q}_h)$.

In the following, we will define PDD and its specifications on model disagreement.

**Definition 1** (Post-deployment deterioration, PDD). *Denote $g$ and $g'$ as ground truth labeling functions in the training and deployed distributions $\boldsymbol{P}$ and $\boldsymbol{Q}$. We say that PDD has occurred when:*

$$\text{err}(f; \boldsymbol{Q}_{g'}) > \text{err}(f; \boldsymbol{P}_g) \tag{1}$$

Intuitively, Eq. (1) suggests that PDD occurs when a model $f$ experiences higher error during deployment. Due to the unsupervised nature of the deployment dataset, PDD monitoring is impossible for any arbitrary $g' \neq g$. Therefore, further assumptions are required to describe a provable setup. To this end, Def. 2 introduces a new and practical concept—model disagreement-based PDD—equivalent to PDD under specific assumptions.

**Definition 2** (Disagreement based PDD (D-PDD)). *We say that D-PDD has occurred when the following holds for some $\epsilon_f < 1$:*

$$\exists h \in \mathcal{H} \quad s.t. \quad \text{err}(h; \boldsymbol{P}_g) \leq \epsilon_f \text{ and } \text{err}(f; \boldsymbol{P}_g) \leq \epsilon_f \text{ and } \text{err}(h; \boldsymbol{Q}_f) > \text{err}(h; \boldsymbol{P}_f) \tag{2}$$

D-PDD in Def. 2 is defined as the situation where there exists an auxiliary model $h \in \mathcal{H}$ achieving equally good performance on $\boldsymbol{P}$ (with a small error $\epsilon_f$) but exhibits strong disagreement with $f$ in $\boldsymbol{Q}$. In this case, the distribution $\boldsymbol{Q}$ is further referred to as a **deteriorating shift**. In the following lemma, we demonstrate the conditions for the equivalence of PDD and D-PDD. As this equivalence happens in probability, Def. 2 allows for certain false positive errors w.r.t. to Def. 1.

**Lemma 2.1** (Equivalence condition). *If we assume (1) that the ground truth labeling functions in the training and deployment distributions are identical ($g = g'$), and (2) that the TV distance between the marginal training and deployment distributions is constrained by some $\kappa$, $TV(\boldsymbol{P}_x, \boldsymbol{Q}_x) \leq \kappa$, then with probability $1 - 2\epsilon_f - \kappa$, PDD is equivalent to D-PDD.*

**Benefits of D-PDD.** Building off Lemma 2.1, the D-PDD framework offers a principled and operationally straightforward approach to monitoring performance in downstream distributions. In contrast, the traditional distribution shift literature (Sugiyama et al., 2007) often overlooks the implications of shifts by detecting them without considering the performance evaluation of the deployed models.

**Figure 2:** PDD, D-PDD, and D-PDDM.

**Algorithmic design.** Tracking D-PDD in finite samples as formulated requires training data. To circumvent this, we **decouple** the detection of D-PDD into two pre-training and deployment stages. The pre-training stage finds a subset $\mathcal{H}_p \subset \mathcal{H}$ whose elements satisfy conditions on $\boldsymbol{P}_g$ in Def. 2 as well as approximates $\text{err}(h; \boldsymbol{P}_f)$, while the deployment stage tracks the last inequality. In this way, information from the training data is compressed into $\mathcal{H}_p$ and the approximation of $\text{err}(h; \boldsymbol{P}_f)$.

# 3 DISAGREEMENT-BASED POST-DEPLOYMENT DETERIORATION MONITORING (D-PDDM) ALGORITHM

The approximation of the disagreement threshold $\text{err}(h; \boldsymbol{P}_f)$ for $h \in \mathcal{H}_p$ can be done via its empirical distribution $\Phi$ computed during pre-training. Thus, we present D-PDDM, a monitoring algorithm detecting D-PDD under *finite samples*. Figure 2 contextualizes D-PDDM within the disagreement framework presented in Sec. 2. D-PDDM only requires the deployed model $f$, a subset of the hypothesis space $\mathcal{H}_p$, and the distribution of the disagreement thresholds $\Phi$ during deployment. The algorithmic protocol is described in two steps:

**1. Pre-training in $\boldsymbol{P}$** Given training data $\mathcal{D}^n$, a base model $f$ trained on $\mathcal{D}^n$, an error tolerance $\epsilon$, and the hypothesis class $\mathcal{H}$, we formulate the subset of hypotheses of interest $\mathcal{H}_p = \{h \in \mathcal{H}; \text{err}(h; \boldsymbol{P}_g) \leq \epsilon\}$. Then, for multiple rounds, the maximum empirical disagreement rate $\max_{h \in \mathcal{H}_p} \widehat{\text{err}}(h; \mathcal{D}^m)$ achievable for independent samples $\mathcal{D}^m$ is appended to $\Phi$.

---

**Algorithm 1** Pre-training

**Require:** $\mathcal{D}^n \sim \boldsymbol{P}_g$, $f$, $\epsilon$, $\mathcal{H}$
1: Train a sub hypothesis space $\mathcal{H}_p := \{h \in \mathcal{H}; \widehat{\text{err}}(h; \boldsymbol{P}_g) \leq \epsilon\}$
2: $\Phi \leftarrow []$
3: **for** $i \leftarrow 1, 2, \ldots$ **do**
4:     $\mathcal{D}^m \sim \boldsymbol{P}_f$
5:     $h \leftarrow \underset{h \in \mathcal{H}_p}{\text{argmax}}\ \widehat{\text{err}}(h; \mathcal{D}^m)$
6:     **append** $\widehat{\text{err}}(h; \mathcal{D}^m)$ to $\Phi$
7: **end for**
8: **return** $\Phi$, $\mathcal{H}_p$

---

**Algorithm 2** D-PDDM test

**Require:** $\mathcal{H}_p$, $\Phi$, $f$, $\alpha$
1: $\mathcal{D}^m \sim \boldsymbol{Q}_f$
2: $h \leftarrow \underset{h \in \mathcal{H}_p}{\text{argmax}}\ \widehat{\text{err}}(h; \mathcal{D}^m)$
3: **return** $\widehat{\text{err}}(h; \mathcal{D}^m) > (1 - \alpha)$ quantile of $\Phi$

---

**2. D-PDDM test in $\boldsymbol{Q}$** Given $\Phi$ and $\mathcal{H}_p$, we approximate the maximal disagreement with $f$ on $\boldsymbol{Q}$: $\text{dis}_Q = \max_{h \in \mathcal{H}_p} \text{err}(h; \boldsymbol{Q}_f)$.

We say D-PDD happens when $\text{dis}_Q$ lies in the top $\alpha$ of $\Phi$.

**Practical considerations** Adopting the Bayesian framework, one can view $\mathcal{H}_p$ as encoding the model's posterior parameter distribution. In this way, one can approximate lines 5 in Algorithm 1 and 2 in Algorithms 2 via computing disagreement rates on weights sampled from the posterior. Appendix B.1 provides a description of the Bayesian perspective and the sampling scheme required to approximate the pre-training and the D-PDDM test algorithm.

As stated previously, the novelty of the decoupling of Def. 2 is what allows D-PDDM to drop the requirement for training data during deployment. On the other hand, all standard baselines including disagreement and distribution shift detection methods rely on computing statistics on the training and testing distributions, which can often not be done efficiently for large and high-dimensional training sets.

# 4 PROVABLE GUARANTEES OF D-PDDM

In this section, we present theoretical guarantees for the proposed D-PDDM algorithm. Recall that the algorithm is tracking a sufficient condition of D-PDD. We first show that with enough samples, when there is non-deteriorating shift, the algorithm achieves low false positive rates with high probability. Then, we show that with enough samples, when there is deteriorating shift, the algorithm provably succeeds. Finally, we discuss pathological cases where the test fails irrespective of sample size. Before stating the theorems we define the following quantities.

## 4.1 PRELIMINARY QUANTITIES

**Definition 3** (Deployed classifier error). *This quantifies the generalization error of the deployed base classifier $f$. This is measured on the distribution seen during training $\boldsymbol{P}_g$,*

$$\epsilon_f := \text{err}(f; \boldsymbol{P}_g) \tag{3}$$

Indeed in Def. 2, we want the population error to be at most $\epsilon_f$, which results in the constraint for the empirical error in the optimization problems of Algorithm 1 at most $\epsilon = \epsilon_f - \epsilon_0$, where $\epsilon_0$ is a hyper-parameter to measure the gap between the empirical and population error.

We also define the VC dimensions of the hypothesis space $\mathcal{H}$ and the subset of interest $\mathcal{H}_p$ as:

$$\mathcal{H}_p := \{h \in \mathcal{H} : \mathrm{err}(h; \boldsymbol{P}_g) \leq \epsilon_f\}, \quad d_p := \mathrm{VC}(\mathcal{H}_p), \quad d := \mathrm{VC}(\mathcal{H})$$

Note that $d_p \leq d$. If the base classifier $f$ is well-trained ($\epsilon_f$ is low), then $d_p$ can be much smaller than $d$ i.e., $d_p \ll d$.

**Definition 4** ($\epsilon_p, \epsilon_q$ maximum error in $\mathcal{H}_p$). *We define the maximum error in $\mathcal{H}_p$ for both $\boldsymbol{P}$ and $\boldsymbol{Q}$ using pseudo-labels from $f$,*

$$\epsilon_p = \max_{h \in \mathcal{H}_p} \mathrm{err}(h; \boldsymbol{P}_f), \quad \epsilon_q = \max_{h \in \mathcal{H}_p} \mathrm{err}(h; \boldsymbol{Q}_f) \tag{4}$$

*Note that empirical quantities of these are also the maximum empirical disagreement rates used in Algo. 1 and Algo. 2. Effectively, the algorithm detects $\epsilon_q - \epsilon_p > 0$ with finite samples.*

**Definition 5** ($\xi$ quantifies D-PDD). *We define $\xi$ to quantify the degree of D-PDD. We adopt Def. 2 and define $\xi$ as*

$$\xi := \max_{h \in \mathcal{H}_p} \{\mathrm{err}(h; \boldsymbol{Q}_f) - \mathrm{err}(h; \boldsymbol{P}_f)\} \tag{5}$$

*Therefore, D-PDDM detects whether $\xi > 0$. Furthermore, $\xi$ is non-negative since $f \in \mathcal{H}_p$. Hence, in case of non-deteriorating shift, $\xi = 0$.*

Note that $\xi \geq \epsilon_q - \epsilon_p$. It follows that $\epsilon_q - \epsilon_p > 0 \implies \xi > 0$, though the reverse implication is not necessarily true. Therefore Algo. 2, ($\epsilon_q - \epsilon_p > 0$) is detecting a sufficient condition of D-PDD ($\xi > 0$).

Next, we relate the amount of D-PDD, $\xi$, with the amount of distribution shift in the form of TV-distance between $\boldsymbol{P}_x$ and $\boldsymbol{Q}_x$. As seen in the Eq. 5, deterioration depends on the complexity of the function class and $\epsilon_f$ which affects the size of $\mathcal{H}_p$. We capture these factors by introducing a mixture distribution $\boldsymbol{U}$:

$$\boldsymbol{U} = \frac{1}{2}\left(\boldsymbol{P}_f + \boldsymbol{Q}_{1-f}\right) \tag{6}$$

**Definition 6** ($\eta$ error gap between $\mathcal{H}_p$ and Bayes optimal). *For the distribution $\boldsymbol{U}$, the gap in error between the best classifier $h \in \mathcal{H}_p$ in the function class and the Bayes optimal classifier is $\eta$:*

$$\eta := \min_{h \in \mathcal{H}_p} \mathrm{err}(h; \boldsymbol{U}) - \mathrm{err}(f_{bayes}; \boldsymbol{U}) \tag{7}$$

Note that $\eta$ depends on the shift and complexity of the function class. We relate various definitions introduced in this section as follows.

**Proposition 4.1** (D-PDD and TV distance). *The relations between $\xi$ (in Def. 5), $\eta$ (in Def. 6), and $\epsilon_p, \epsilon_q$ (in Def. 3 and 4) are as follows:*

$$\xi = \mathrm{TV} - 2\eta \geq 0 \tag{8}$$
$$\xi \geq \epsilon_q - \epsilon_p \geq \xi - 2\epsilon_f \tag{9}$$

We denote the total variation distance between $\boldsymbol{P}_x$ and $\boldsymbol{Q}_x$ as TV. Intuitively, D-PDD is defined in such a way that after deployment, if we are uncertain of the performance of $f$, then the shift is deteriorating. In general, for simpler function classes such as linear models, by looking at one region of the domain ($\boldsymbol{P}_x$) it may be possible to be certain about the performance of another region ($\boldsymbol{Q}_x$), this is captured in Eq. 8. For very complex function classes, $\eta$ can be low, hence $\xi > 0$ for most shifts. For simple function classes, $\eta$ can be high, in which case $\xi$ may not be positive and hence a non-deteriorating shift. This thus highlights a trade-off with selecting expressive functions to capture complex patterns in the data.

### 4.2 D-PDDM ALGORITHM IN NON-DETERIORATING SHIFT

D-PDDM aims to monitor and detect D-PDD in finite samples, which can inherently lead to false positives (FPR) when the shift is non-deteriorating. Therefore we set a tolerance factor $\alpha$ in Alg. 2 to account for the test's robustness. In this subsection, we show that for D-PDDM, the FPR of the detection can be close to $\alpha$ for *any* shift in the data distribution. Furthermore, we show that the FPR

can also be less than $\alpha$ in some cases. Specifically, in the case of non deteriorating shift by Def. 2, the following holds:

$$\forall h \in \mathcal{H}_p : \ \mathrm{err}(h; \boldsymbol{Q}_f) \leq \mathrm{err}(h; \boldsymbol{P}_f) \implies \epsilon_q \leq \epsilon_p \tag{10}$$

Note that D-PDDM intuitively detects whether $\epsilon_q > \epsilon_p$. Since the above equation shows that $\epsilon_q \leq \epsilon_p$, given enough samples, the test will succeed. Recall that $n$ is the number of samples given from $\boldsymbol{P}_g$ and $m$ is the number of samples required from $\boldsymbol{Q}_x$. In the theorem below, the significance level $\alpha$ refers to the desired FPR.

**Theorem 4.2.** *For $\gamma \leq \alpha$, when there is no deteriorating shift (no D-PDD) in Eq. 10, for a chosen significance level of $\alpha$, the FPR of D-PDDM is at most $\gamma + (1 - \gamma)\, \mathcal{O}\left(\exp\left(-n\epsilon_0^2 + d\right)\right)$ if*

$$m \in \mathcal{O}\left( \left( \frac{1 - \sqrt{\delta}}{\epsilon_p - \epsilon_q} \right)^2 \left( d_p + \ln \frac{1}{\gamma} \right) \right) \tag{11}$$

*and $\epsilon_p - \epsilon_q > 0$, where $\delta = (d_p + \ln \frac{1}{\alpha})/(d_p + \ln \frac{1}{\gamma})$.*

In the case of non deteriorating shifts (specifically $\epsilon_p > \epsilon_q$) the FPR may be even less than $\alpha$ given that $m$ and $n$ are sufficiently large. The more samples from $\boldsymbol{Q}_x$ we have, the lesser the FPR in these cases. For any general case, by setting $\gamma = \alpha$ (i.e., $\delta = 1$) in the above theorem, we immediately have:

**Corollary 4.3.** *For a chosen significance level $\alpha$, the FPR of D-PDDM (Alg. 2) is no more than $\alpha + (1 - \alpha)\, \mathcal{O}\left(\exp\left(-n\epsilon_0^2 + d\right)\right)$.*

**Practical insights.** The corollary asserts the robustness of D-PDDM against unnecessarily flagging non-deteriorating shifts. Independent of the number of deployment samples $m$, for any given significance level $\alpha$, the FPR is only slightly worse, with the additive term decaying exponentially in the number of training samples. For many practical ML pipelines that by-and-large employ linear and forest models among others of manageable VC-dimension, having these guarantees means that a D-PDDM audit likely won't negatively impact the continuity and quality of service.

## 4.3 D-PDDM ALGORITHM IN DETERIORATING SHIFT

When deteriorating shift occurs:

$$\exists h \in \mathcal{H}_p : \mathrm{err}(h; \boldsymbol{Q}_f) > \mathrm{err}(h; \boldsymbol{P}_f) \tag{12}$$

However, this does not necessarily imply that $\epsilon_q > \epsilon_p$ which is ultimately the condition monitored by D-PDDM. In the following, we break down the possible scenarios.

**Regime 1. Deteriorating shift and $\epsilon_q > \epsilon_p$.** In this case, Theorem 4.4 demonstrates that the D-PDDM algorithm detects deteriorating shift with provable high TPR. Here, the significance level $\alpha$ is understood to be 1 minus the desired TPR.

**Theorem 4.4.** *For $\beta > 0$, when deteriorating shift occurs, for a chosen significance level of $\alpha$, the TPR of D-PDDM (Alg. 2) is at least $(1 - \beta)\left(1 - \mathcal{O}\left(\exp\left(-n\epsilon_0^2 + d\right)\right)\right)$ if*

$$m \in \mathcal{O}\left( \left( \frac{1 + \sqrt{\delta}}{\xi - 2\epsilon_f} \right)^2 \left( d_p + \ln \frac{1}{\beta} \right) \right) \tag{13}$$

*and $\epsilon_q - \epsilon_p > 0$, where $\delta = (d_p + \ln \frac{1}{\alpha})/(d_p + \ln \frac{1}{\beta})$.*

Notably, $\xi$ in the denominator indicates that as the shift becomes more deteriorating, D-PDDM requires fewer samples $m$ to detect, evidencing its effectiveness. Also, having a high-quality base classifier $f$ with low $\epsilon_f$ is much better for the detection: this is seen through in Eq. 9 where low $\epsilon_f$ makes the monitoring more faithful. Another remark is that $m$ depends on $d_p$ which can be much less than $d$ with $n$ being dependent on the latter. The test, thus, can work for a $m$ significantly smaller than $n$. The dependency on $n$ is due to the requirement of satisfaction of the first condition in Def. 2. In the constrained optimization problems in Algo. 1, the constraint is satisfied but the population constraint will be satisfied either for larger $\epsilon_0$ or for sufficiently large $n$ as seen in the theorem.

**Regime 2. (Possible tradeoff)** -Deteriorating shift but $\epsilon_q \leq \epsilon_p$. In this case, Theorem 4.5 demonstrates that either the false negative or false positive rates (FNR, FPR) should be high. The illustration in Fig. 3 exemplifies this failure mode, and how a low $\epsilon_f$ can help alleviate it.

**Theorem 4.5.** *When deteriorating shift occurs and $\epsilon_q \leq \epsilon_p$, for a chosen significance level of $\alpha$, the TPR of Alg. 2 is $\mathcal{O}(\alpha)$.*

If $\alpha$ is low, then by Theorem 4.5 we have that the FNR is high. On the other hand, if $\alpha$ is high, then by Corollary 4.3 we have that the FPR can be very high, thereby trading off the significance level of D-PDDM to reduce FNR but loosening guarantees on FPRs.

### 4.3.1 SOLUTIONS FOR FNR/FPR TRADEOFF

This part provides a possible failure scenario illustrated in Fig. 3. Notably, we will highlight a badly trained base classifier $f$ in the possible failure scenario (Fig. 3 (a)), if $f$ is trained with lower $\epsilon_f$, can move to the scenarios (Fig. 3 (b) and (c)) where the D-PDDM algorithm can succeed.

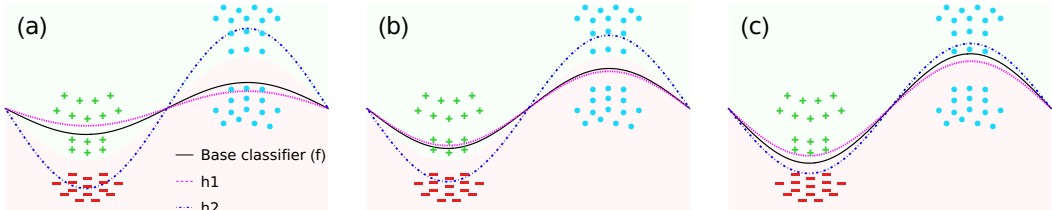

**Figure 3: Illustration of the FNR/FPR tradeoff and its remedy.** The background color indicates the fixed ground truth. Positive and Negative points are from $\boldsymbol{P}_g$ (labeled) and the unlabeled points are from $\boldsymbol{Q}_x$. The solid black curve represents the deployed base classifier $f$. The dotted Pink ($h_1$) and Blue ($h_2$) curves represent the envelope boundary for $\mathcal{H}_p$ i.e., all the functions passing between these two curves are contained in $\mathcal{H}_p$. (a) Failure scenario (i.e, Regime 2) where D-PDDM algorithm fails. (b) No deteriorating shift scenario. (c) Deteriorating shift and the D-PDDM algorithm succeeds. In summery, a decreasing on $\epsilon_f$ could move the failure scenario to the solvable scenarios (a) or (b).

In Fig. 3 (a), if $f$ is not well-trained, we will encounter a failure scenario. The disagreement of $h_1$ with $f$ on $\boldsymbol{Q}_x$ is larger than that of $\boldsymbol{P}_x$, evidencing D-PDD. However, note that $h_2$ can maximize $\epsilon_p$ more than any function (in $\mathcal{H}_p$) can maximize $\epsilon_q$, which implies $\epsilon_p > \epsilon_q$. If $f$ is better trained in Fig. 3 (b), for all functions in $\mathcal{H}_p$ (curves between $h_1$ and $h_2$) disagreement with $f$ on $\boldsymbol{P}_x$ is not less than that of $\boldsymbol{Q}_x$. Hence there is no deteriorating shift and D-PDDM algorithm could provably address this. Alternatively, if $f$ is trained well and is closest to the ground truth Fig. 3 (c) the disagreement of $h_2$ with $f$ on $\boldsymbol{Q}_x$ is more than that of $\boldsymbol{P}_x$. Also, note that $\epsilon_p = 0$ since there is no function that can have any error on $\boldsymbol{P}_f$. However, $h_2$ can be the classifier to get non-zero $\epsilon_q$ which gives $\epsilon_q > \epsilon_p$. Hence (c) recovers the Regime 1 and is solvable.

**Practical implications.** Training base classifiers with strong in-distribution generalization performance helps in reducing the likelihood of falling into **Regime 2**. Then, Theorem 4.5 guarantees that with high probability, the desired TPR of D-PDDM can be achieved modulo an exponentially decaying factor in the number of training samples. In this way, D-PDDM is robust in monitoring deteriorating shifts with provable TPR guarantees, satisfying the robustness desiderata for PDD monitoring.

## 5 EXPERIMENTS

Our experiments done on synthetic and real-world vision and large-scale healthcare datasets benchmark D-PDDM[1] on *deteriorating* and *non-deteriorating shifts* against other competitive baselines.

---

[1]An implementation of D-PDDM can be found here.

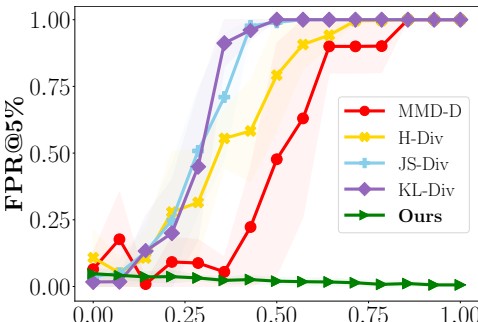

**Figure 4:** Non deteriorating shift in synthetic data.

| | CIFAR 10.1 | | |
|---|---|---|---|
| Unlabeled test size | 50 | 100 | 200 |
| MMD-D | $0.26 \pm 0.05$ | $0.53 \pm 0.11$ | $0.93 \pm 0.04$ |
| H-divergence | $0.23 \pm 0.05$ | $0.45 \pm 0.05$ | $0.78 \pm 0.05$ |
| JS-divergence | $0.05 \pm 0.03$ | $0.09 \pm 0.03$ | $0.24 \pm 0.05$ |
| KL-divergence | $0.16 \pm 0.05$ | $0.38 \pm 0.06$ | $0.75 \pm 0.04$ |
| **D-PDDM (Ours)** | $\mathbf{0.55 \pm 0.05}$ | $\mathbf{0.71 \pm 0.05}$ | $\mathbf{0.93 \pm 0.03}$ |

**Table 2:** True Positive Rate ($\alpha = 0.05$) on CIFAR10.

## 5.1 EXPERIMENTAL SETUP

**Dataset** *Synthetic data*. (1) Synthetic data is generated based on a sinusoidal hypersurface partitioning the feature space into two halves and assigning positive or negative labels accordingly. For details on the data generation process as well as deteriorating and non-deteriorating shift induction, see Appendix B.2. All experiments use $n = 10,000$ when sampling in-distribution and $m = 4,000$ when sampling on the shifted distributions. *Benchmark and real-world hospital data* (2) CIFAR-10.1 dataset (Recht et al., 2019) where shift comes from subtle changes in the dataset creation process; and (3) the General Medicine INpatient Initiative (GEMINI) dataset (Waters et al., 2023; Verma et al., 2021), which collects and standardizes large-scale administrative and clinical data from hospitals.

**Implementation & Baselines.** In all of our implementations, our hypothesis class is the space of neural networks restricted to several layers of $\approx 32$ hidden nodes each to respect the expressivity constraints of our analysis. To demonstrate that our test enjoys low FPR on non deteriorating shifts and high TPR on deteriorating shifts, we compare it against several distribution divergence-based detection methods from the literature: Deep Kernel MMD (MMD-D) (Liu et al., 2020), H-divergence (Zhao et al., 2022), adapt several $f$-divergences (Acuna et al., 2021) into a hypothesis test via permutation testing (Ernst, 2004), Black Box Shift Detection (BBSD) (Lipton et al., 2018), and Relative Mahalanobis Distance (RMD) Ren et al. (2021). Details can be found in Appendix B.5.

**Evaluations.** *Synthetic data*. For non-deteriorating shifts, we report the FPR at level $\alpha = 0.05$ of our method and the baselines. We run 500 permutations times 100 independent tests for each baseline whereas for D-PDDM, we report 100 independent realizations to compute the TPR/FPR, each run running 500 pre-training steps. Importantly, the baseline methods have oracle access to the generating distributions $P_x$ and $Q_x$ and re-sample new sets of data from $P_x$ and $Q_x$ for each permutation, ensuring a fair comparison with our theoretical algorithm and empowers the baselines as it de-biases their test statistics away from one particular sampling of $P_x$ and $Q_x$, as they would otherwise have to sample with replacement. *Real-world dataset evaluation.* For CIFAR 10.1 where there is known post-deployment deterioration, we evaluate the baselines' and D-PDDM's ability to detect shift at level $\alpha = 0.05$ in $\{50, 100, 200\}$-shot scenarios. For the GEMINI health dataset, we study the detection rates of said models on temporally-split sub-datasets and mixtures of subpopulation splits incurring deteriorating changes. Additional information for the Gemini Dataset and splits is present in Appendix. B.6.

## 5.2 RESULTS & ANALYSIS

**Synthetic data.** We quantify the amount of non-deteriorating shift using a gap parameter $\Delta$ between $[0, 1]$ which effectively stretches the distributions of features away from the true decision boundary. We observe that the baselines eventually achieve an FPR of 1.0 with high certainty, while D-PDDM using a base classifier with in-distribution generalization error 0.1 achieves superior robustness to non-deteriorating shifts with FPR much lower than $\alpha = 5\%$. Given that the baselines essentially use some notion of distance either in $d$-dimensional feature space or in a learned space (as in the case of MMD-D) as a test statistic and perform a permutation test, it stands to reason that these methods pick up on the slightest changes in the distribution.

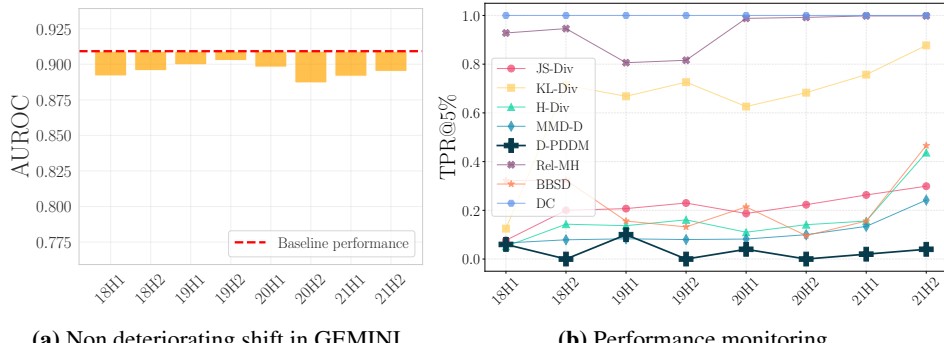

**(a)** Non deteriorating shift in GEMINI   **(b)** Performance monitoring

**Figure 5:** Performances in time evolving shifted test data from GEMINI. **(a)** Performance drop (bar plot) small, thereby a non deteriorating temporal shift. **(b)** Time evolving shift monitoring. D-PDDM is robust with small False Positive Rate (FPR) at level $\alpha = 0.05$.

**CIFAR10.1.** CIFAR10.1 is known to have strong deteriorating shifts w.r.t. CIFAR10 due to its curation. We observe that D-PDDM is competitive with respect to the baselines. In particular, for each few-shot setting, D-PDDM enjoys higher TPR at level $\alpha = 0.05$. Importantly, we remark that even when benchmarked against divergence baselines that flag any changes in the distribution of features, D-PDDM still evaluates better. This finding empirically suggests that using disagreement rate as the test statistic, while explicitly accounting for in-distribution performance (via optimizing over $\mathcal{H}_p$) and deployment distribution performance (via maximizing the disagreement objective in Algorithm 2), implicitly attends to shifts in the features of the data as well through the ease or difficulty of fitting the disagreement objective using $h \in \mathcal{H}_p$.

**GEMINI temporal shift.** On GEMINI, we first train the base classifier $f$ on data prior to 2018. $f$ is then deployed on dataset splits corresponding to subsequent half-years. In Fig. 5(a), we observe that there is little to no apparent trend in performance degradation across time, thus it could be understood that this temporal data shift is non-deteriorating. Viewed this way, an ideal monitoring algorithm should resist flagging the ML system, allowing its continual performance. Indeed, in Fig. 5(b), D-PDDM is least reactive to detection while $f$-divergence and H-divergence baselines unnecessarily alert the system of shifts, achieving false positive rates above $0.1$ consistently across temporal shifts.

**GEMINI age shift.** We further manufacture deteriorating shifts by training the model on data from adults between 18 and 52 years old, and assessing its performance on mixtures of deployment datasets containing varying proportions of unseen data from (i) training distribution and (ii) adults above 85. In Fig. 6(a), we observe clear post-deployment deterioration as we introduce more data from the second group, and would hope that monitoring algorithms flag these deployments. Indeed, as illustrated by Fig. 6(b), all methods properly detect this drastic change. Notable, D-PDDM outperforms most baselines, staying competitive with the highest TPRs at all proportion of mixutres of data. This demonstrates that D-PDDM attents to deteriorating chagnes in the distribution of features as fast as any baseline method.

# 6   RELATED WORK

**Performance monitoring of ML models & deteriorating shift.** Evaluating a model's reliability during deployment is crucial for the safety and effectiveness of the machine learning pipeline over time. For example, Feng et al. (2024b;a) provided a causal viewpoint wherein the challenge to adapt to diverse scenarios still remain due to the lack of access to the true causal graph. Model disagreement is often used as a monitoring tool for the model generalization (Jiang et al., 2022; Chuang et al., 2020; Ginsberg et al., 2023; Rosenfeld & Garg, 2023). Our paper significantly differs from these works via our theoretical analysis of D-PDDM via guarantees on the FPR and TPR, whereas these works provided sufficient conditions in either i.i.d. or various shift scenarios. Several works in the recent literature differentiate shifts in terms of deteriorating or non-deteriorating shifts. Podkopaev & Ramdas (2021) studied deteriorating shift detection in the continuous monitoring setting using a sequential hypothesis test. Due to the setting being sequential in nature, their method requires true

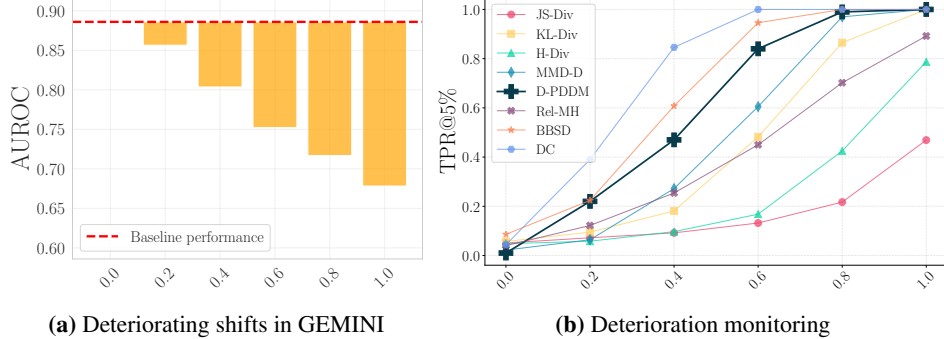

(a) Deteriorating shifts in GEMINI      (b) Deterioration monitoring

**Figure 6:** Monitoring results on artificially shifted test data from hospital (GEMINI). **(a)** Performance drop (bar plot) is significant when the degree of shift is large ($0.0 \rightarrow 1.0$) **(b)** Results on different monitoring methods, P-PDDM has a better True Positive Rate (TPR) at level $\alpha = 0.05$.

labels from $\boldsymbol{Q}$ immediately after prediction or at the least in a delayed fashion. Our setting deviates from theirs in the sense that labels from $\boldsymbol{Q}$ are not available at any time. Other related empirical works along this literature are Wang et al. (2023); Kamulete (2022).

**Distribution shift detection.** Methods to detect distribution shift arise from different perspectives. In covariate shift detection, (Lopez-Paz & Oquab, 2016; Liu et al., 2020; Zhao et al., 2022) treated detection as two-sample tests via classifier, Deep Kernel MMD, and H-divergence. For label shift on the other hand, (Lipton et al., 2018; Azizzadenesheli et al., 2019) formulated the problem as a convex optimization problem by solving the label distribution ratio $\alpha = \boldsymbol{Q}(y)/\boldsymbol{P}(y)$. The problem of (out-of-distribution) OOD (Liang et al., 2017; Kamulete, 2022) detection seeks to detect if an individual sample $x$ comes from the training distribution $x \sim \boldsymbol{P}(x)$. Some previous works (Ren et al., 2019; Morningstar et al., 2021) also adopted the methods in covariate shift detection and generalization by estimating the density ratio for the identification of OOD samples. Whilst these methods detect shifts, they are constrained by their requirement of training data post-deployment and do not consider the extent to which shifts affect model performance.

**Estimating test error with unlabeled data.** Another rich body of research is the estimation of test error. This technique and its variants are often inspired by domain adaptation theories (Ben-David et al., 2006; 2010; Acuna et al., 2021; Ganin et al., 2016), seeking guarantess in the form of $\text{err}(f; \boldsymbol{Q}_g) \leq \text{err}(f; \boldsymbol{P}_g) + \Delta(f, \mathcal{H})$, with $\Delta(f, \mathcal{H}) = \sup_{h \in \mathcal{H}} |\text{err}(h; \boldsymbol{P}_f) - \text{err}(h; \boldsymbol{Q}_f)|$. This objective can be alternatively viewed as searching for a critic function $h \in \mathcal{H}$ to maximize the performance gap (Rosenfeld & Garg, 2023; Jiang et al., 2021). One could thus provably estimate the upper bound of the test distribution error. These theories, however, implicitly assume the availability of training data. Further, they assume that the test error should be larger than the training error, making them sensitive to non deteriorating shifts as well i.e., high FPR in detection. Our theory for D-PDDM encompasses this regime of change as well.

## 7 CONCLUSION

We study the problem of post-deployment deterioration monitoring of machine learning models in the setting where labels from test distribution are unavailable. We propose a two-stage disagreement-based monitoring algorithm, D-PDDM, which monitors and detects deteriorating changes in the deployment dataset while being resilient to flagging non-deteriorating changes. Importantly, our method does not require any training data during monitoring, allowing for efficient out-of-the-box deployment in many machine learning pipelines across various domains. We provide statistical guarantees for low FPR in the case of non-deteriorating shifts and reliable TPR in the deteriorating shift. Empirically, we validate insights from our theory on various synthetic and real-world vision and healthcare datasets evidencing the effective use of D-PDDM. The empirical success of D-PDDM signals a step toward the *robust, scalable, and efficient* deployment of mechanisms to audit and monitor machine learning pipelines in the break of dawn of ubiquitous AI.

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
