# A APPENDIX

## A.1 PROOFS IN SEC. 2

**Lemma A.1** (Equivalence condition). *If we assume (1) identical ground truth labeling function in the training and deployment $g = g'$, (2) Restricted TV distance between training and deployment $TV(\boldsymbol{P}_x, \boldsymbol{Q}_x) \leq \kappa$, then with probability $1 - 2\epsilon - \kappa$, PDD is equivalent to D-PDD.*

*Proof.* **Step 1:** Def 1 $\rightarrow$ Def 2. If $g' = g$, it is clear

$$\text{err}(f, \boldsymbol{Q}_g) > \text{err}(f, \boldsymbol{P}_g)$$

Then we set $h = g \in \mathcal{H}$, we have Def 2.
**Step 2:** Def 2 $\rightarrow$ Def 1. If $g' = g$, we need to prove

$$\text{err}(f, \boldsymbol{Q}_g) > \text{err}(f, \boldsymbol{P}_g)$$

Given the disagreement condition in Def 2,

$$\text{err}(f, \boldsymbol{Q}_h) > \text{err}(f, \boldsymbol{P}_h)$$

We need to demonstrate $h = g$ with high probability, given the binary risk definition and Markov inequality, we have high probability $1 - \epsilon$ such that:

$$h = g, \quad f = g$$

Therefore $\text{err}(f, \boldsymbol{P}_g) = \text{err}(f, \boldsymbol{P}_h)$ happens in probability $\boldsymbol{P}$ with $1 - \epsilon$. Then we consider this events in $\boldsymbol{Q}$, given a small TV-distance ($\kappa$) between $\boldsymbol{P}$ and $\boldsymbol{Q}$, we have

$$|\boldsymbol{P}(f(x) = g(x)) - \boldsymbol{Q}(f(x) = g(x))| \leq \text{TV}(\boldsymbol{P}_x, \boldsymbol{Q}_x) \leq \kappa$$

Thus with high probability $1 - \kappa - \epsilon$ in $\boldsymbol{Q}$, we still have $f(x) = g(x)$. Union bounding yields the desired conclusion. $\square$

## A.2 PROOFS IN SEC. 4

**Lemma A.2.** *For any $\gamma > 0$, $\mu > \epsilon_q$, we have $\widehat{\text{err}}(h; \boldsymbol{Q}_f) \leq \mu$ for all $h \in \mathcal{H}_p$ with probability at least $1 - \gamma$ if*

$$m \in \mathcal{O}\left(\frac{d_p + \ln\frac{1}{\gamma}}{(\mu - \epsilon_q)^2}\right) \tag{14}$$

*Proof.* We use the generalization bound for agnostic learning in Shalev-Shwartz & Ben-David (2014).

$$ce^{d_p}e^{-\epsilon^2 m} \geq \Pr_{X,Y \sim \boldsymbol{Q}_{1-f}^m}[\exists h \in \mathcal{H}_p : \text{err}(h; \boldsymbol{Q}_{1-f}) - \widehat{\text{err}}(h; \boldsymbol{Q}_{1-f}) \geq \epsilon] \tag{15}$$

$$= \Pr_{X,Y \sim \boldsymbol{Q}_{1-f}^m}[\exists h \in \mathcal{H}_p : \widehat{\text{err}}(h; \boldsymbol{Q}_f) \geq \text{err}(h; \boldsymbol{Q}_f) + \epsilon] \tag{16}$$

$$\geq \Pr_{X,Y \sim \boldsymbol{Q}_{1-f}^m}[\exists h \in \mathcal{H}_p : \widehat{\text{err}}(h; \boldsymbol{Q}_f) \geq \epsilon_q + \epsilon] \tag{17}$$

Choose $\epsilon = \mu - \epsilon_q$ for any $\mu > \epsilon_q$. Now,

$$ce^{d_p}e^{-\epsilon^2 m} \leq \gamma \tag{18}$$

$$m \in \mathcal{O}\left(\frac{d_p + \ln\frac{1}{\gamma}}{(\mu - \epsilon_q)^2}\right) \tag{19}$$

$\square$

**Lemma A.3.** *For any $h \in \mathcal{H}$, if the $\widehat{\text{err}}(h; \boldsymbol{P}_f) \leq \epsilon_f - \epsilon_0$ then with probability at least $1 - \mathcal{O}\left(\exp\left(-n\epsilon_0^2 + d\right)\right)$, $h$ will be in $\mathcal{H}_p$*

*Proof.* We use the generalization bound for agnostic learning in Shalev-Shwartz & Ben-David (2014).

$$ce^d e^{-\epsilon^2 n} \geq \Pr_{X,Y \sim \boldsymbol{P}_g^n} [\exists h \in \mathcal{H} : \text{err}(h; \boldsymbol{P}_g) - \widehat{\text{err}}(h; \boldsymbol{P}_g) \geq \epsilon] \tag{20}$$

$$= \Pr_{X,Y \sim \boldsymbol{P}_g^n} [\exists h \in \mathcal{H} : \text{err}(h; \boldsymbol{P}_g) \geq \widehat{\text{err}}(h; \boldsymbol{P}_g) + \epsilon] \tag{21}$$

$$\geq \Pr_{X,Y \sim \boldsymbol{P}_g^n} [\exists h \in \mathcal{H} : \text{err}(h; \boldsymbol{P}_g) \geq \epsilon_f - \epsilon_0 + \epsilon] \tag{22}$$

Choose $\epsilon = \epsilon_0$ to get

$$\Pr_{X,Y \sim \boldsymbol{P}_g^n} [\exists h \in \mathcal{H} : \text{err}(h; \boldsymbol{P}_g) \geq \epsilon_f] \leq ce^d e^{-\epsilon^2 n} \tag{23}$$

$\square$

**Theorem A.4.** *For $\gamma \leq \alpha$, when there is no deteriorating shift, for a chosen significance level of $\alpha$, the FPR of Algo. 2 is at most $\gamma + (1 - \gamma) \mathcal{O}\left(\exp\left(-n\epsilon_0^2 + d\right)\right)$ if*

$$m \in \mathcal{O}\left(\left(\frac{1 - \sqrt{\delta}}{\epsilon_p - \epsilon_q}\right)^2 \left(d_p + \ln \frac{1}{\gamma}\right)\right) \tag{24}$$

*and $\epsilon_p - \epsilon_q > 0$, where $\delta = \frac{d_p + \ln \frac{1}{\alpha}}{d_p + \ln \frac{1}{\gamma}}$*

*Proof.* We show that in the case of no deteriorating shift (which implies $\epsilon_p \geq \epsilon_q$) the false positive rate cannot be more than $\alpha$ and also having more samples from $\boldsymbol{Q}_x$ will decrease the false positive rate if $\epsilon_p > \epsilon_q$.

We assume that during pre-training phase, while populating $\Phi$ we discard disagreement from $h \notin \mathcal{H}_p$ i.e., not satisfying the constraint $\text{err}(h; \boldsymbol{P}_f) \leq \epsilon_f$. We cannot do the same during the detection phase since the detection phase is time-sensitive. Due to this, we have to account for $h \notin \mathcal{H}_p$ in the FPR calculation.

Now, FPR can be written and bounded as follows. Let $\mu$ be the disagreement at $1 - \alpha$ percentile of $\Phi$

$$\text{FPR} = \Pr\left[\widehat{\text{err}}(h; \boldsymbol{Q}_f) \geq \mu\right] \tag{25}$$

$$= \Pr\left[\{\{h \notin \mathcal{H}_p\} \wedge \{\widehat{\text{err}}(h; \boldsymbol{Q}_f) \geq \mu\}\} \vee \{\{h \in \mathcal{H}_p\} \wedge \{\widehat{\text{err}}(h; \boldsymbol{Q}_f) \geq \mu\}\}\right] \tag{26}$$

$$\leq \Pr\left[\{h \notin \mathcal{H}_p\} \vee \{\{h \in \mathcal{H}_p\} \wedge \{\widehat{\text{err}}(h; \boldsymbol{Q}_f) \geq \mu\}\}\right] \tag{27}$$

$$\leq \Pr\left[h \notin \mathcal{H}_p\right] + \Pr\left[\{\widehat{\text{err}}(h; \boldsymbol{Q}_f) \geq \mu\} \mid \{h \in \mathcal{H}_p\}\right] \Pr\left[h \in \mathcal{H}_p\right] \tag{28}$$

$$= \gamma + (1 - \gamma) \Pr\left[h \notin \mathcal{H}_p\right] \tag{29}$$

$$\text{FPR} \leq \gamma + (1 - \gamma) \mathcal{O}\left(\exp\left(-n\epsilon_0^2 + d\right)\right) \tag{30}$$

where last equation comes from A.3 and $\gamma := \Pr\left[\{\widehat{\text{err}}(h; \boldsymbol{Q}_f) \geq \mu\} \mid \{h \in \mathcal{H}_p\}\right]$

Now, we derive sample complexity $m$ in terms of $\gamma$. Using A.2 on $\boldsymbol{P}$ with $1 - \alpha$ probability we get

$$m \in \mathcal{O}\left(\frac{d_p + \ln \frac{1}{\alpha}}{(\mu - \epsilon_p)^2}\right) \tag{31}$$

We use $\mu \in \Omega\left(\epsilon_p + \sqrt{\frac{d_p + \ln \frac{1}{\alpha}}{m}}\right)$ from above while using A.2 on $\boldsymbol{Q}$ with $1 - \gamma$ probability to get

$$m \in \mathcal{O}\left(\left(\frac{1 - \sqrt{\frac{d_p + \ln \frac{1}{\alpha}}{d_p + \ln \frac{1}{\gamma}}}}{(\epsilon_p - \epsilon_q)}\right)^2 \left(d_p + \ln \frac{1}{\gamma}\right)\right) \quad \text{for } \gamma < \alpha \tag{32}$$

Note that since the chosen $\mu$ was greater than $\epsilon_p$ and we are dealing with the case $\epsilon_p > \epsilon_q$, we get that the chosen $\mu$ is greater than $\epsilon_q$. Thus the requirement of $\mu$ is satisfied for A.2 while using for $\boldsymbol{Q}$.

$\square$

This theorem shows that when there are non deteriorating shifts (specifically $\epsilon_p > \epsilon_q$) FPR may be even less than $\alpha$, given $m$ and $n$ is sufficiently large. The more samples from $\boldsymbol{Q}_x$ we have the lesser the FPR in these cases. For any general case, by setting $\gamma = \alpha$ (i.e., $\delta = 1$) in the above theorem, we obtain the following:

**Corollary A.5.** *For a chosen significance level $\alpha$, the FPR of the P-PDDM algorithm is no more than $\alpha + (1 - \alpha)\, \mathcal{O}\left(\exp\left(-n\epsilon_0^2 + d\right)\right)$.*

Note that this statement is independent of $m$ and the distribution shift. If $n$ is sufficiently large, the exponential term is small. This is often the case when the base classifier error $\epsilon_f$ is small, which is an indicator that a large number of samples ($n$) were available from $\boldsymbol{P}_g$. Ignoring non deteriorating shift (and $\boldsymbol{Q}_x \neq \boldsymbol{P}_x$) cases while calculating $\Phi$ in Algo. 2 does not adversely affect the FPR of the test.

**Lemma A.6.** *For any $\beta > 0, \mu < \epsilon_q$, there exists an $h \in \mathcal{H}_p$ such that $\widehat{\mathrm{err}}(h; \boldsymbol{Q}_f) \geq \mu$ with probability at least $1 - \beta$ if*

$$m \geq \mathcal{O}\left(\frac{d_q + \ln \frac{1}{\beta}}{(\epsilon_q - \mu)^2}\right) \tag{33}$$

*Proof.* We use the generalization bound for agnostic learning case Shalev-Shwartz & Ben-David (2014).

$$ce^{d_p}e^{-\epsilon^2 m} \geq \Pr_{X,Y\sim\boldsymbol{Q}_{1-f}^m}[\exists h \in \mathcal{H}_p : \widehat{\mathrm{err}}(h; \boldsymbol{Q}_{1-f}) - \mathrm{err}(h; \boldsymbol{Q}_{1-f}) \geq \epsilon] \tag{34}$$

$$= \Pr_{X,Y\sim\boldsymbol{Q}_{1-f}^m}[\exists h \in \mathcal{H}_p : \widehat{\mathrm{err}}(h; \boldsymbol{Q}_f) \leq \mathrm{err}(h; \boldsymbol{Q}_f) - \epsilon] \tag{35}$$

$$\overset{(a)}{=} \Pr_{X,Y\sim\boldsymbol{Q}_{1-f}^m}[\forall h \in \mathcal{H}_p : \widehat{\mathrm{err}}(h; \boldsymbol{Q}_f) \leq \epsilon_q - \epsilon] \tag{36}$$

where (a) follows from Def. 4
Choose $\epsilon = \epsilon_q - \mu$ for any $\mu < \epsilon_q$

$$ce^{d_q}e^{-\epsilon^2 m} \leq \beta \tag{37}$$

$$m \geq \mathcal{O}\left(\frac{d_q + \ln \frac{1}{\beta}}{(\epsilon_q - \mu)^2}\right) \tag{38}$$

$\square$

**Proposition A.7** (D-PDD and TV distance). *The relations between $\xi$ (in Def. 5), $\eta$ (in Def. 6), and $\epsilon_p$, $\epsilon_q$ (in Def. 3 and 4) are as follows:*

$$\xi = \mathrm{TV} - 2\eta \geq 0 \tag{39}$$
$$\xi \geq \epsilon_q - \epsilon_p \geq \xi - 2\epsilon_f \tag{40}$$

*Proof.* Recall the definition of $\boldsymbol{U}$ from 6. We first derive the Bayes error in terms of TV distance. Let

$$A = \{x \in \mathcal{X} \mid \boldsymbol{Q}_x(x) \leq \boldsymbol{P}_x(x)\} \tag{41}$$

$$A' = \{x \in \mathcal{X} \mid \boldsymbol{Q}_x(x) > \boldsymbol{P}_x(x)\} \tag{42}$$

The TV distance is equal to half of the $L_1$ distance. Note that $\boldsymbol{P}_x(A) + \boldsymbol{P}_x(A') = 1$ and similarly for $\boldsymbol{Q}_x$. [2]

$$\mathrm{TV}(\boldsymbol{P}_x, \boldsymbol{Q}_x) = \frac{1}{2}\left(\boldsymbol{P}_x(A) - \boldsymbol{Q}_x(A) + \boldsymbol{Q}_x(A') - \boldsymbol{P}_x(A')\right) \tag{43}$$

$$= 1 - \boldsymbol{P}_x(A') - \boldsymbol{Q}_x(A) \tag{44}$$

---

[2]With some abuse of notation, we use the same notation for both pdf and probability measure.

Now, we use the definition of $U$ and the above TV relation to get the following

$$\text{err}\left(f_{\text{bayes}}; U\right) = \frac{1}{2}\left(\text{err}\left(f_{\text{bayes}}; P_f\right) + \text{err}\left(f_{\text{bayes}}; Q_{1-f}\right)\right) = \frac{1}{2}\left(Q_x(A) + P_x(A')\right) \tag{45}$$

$$= \frac{1}{2}\left(1 - \text{TV}(P_x, Q_x)\right) \tag{46}$$

Next, with the above result and $\eta$ in Eq. 6 we derive Eq. 8

$$\eta + \text{err}(f_{\text{bayes}}; U) = \min_{h \in \mathcal{H}_p} \text{err}(h; U) = \frac{1}{2} \min_{h \in \mathcal{H}_p}\left(\text{err}(h; P_f) + \text{err}(h; Q_{1-f})\right) \tag{47}$$

$$2\eta + 1 - \text{TV} = \min_{h \in \mathcal{H}_p}\left(\text{err}(h; P_f) + \text{err}(h; Q_{1-f})\right) \geq \min_{h \in \mathcal{H}_p} \text{err}(h; Q_{1-f}) \tag{48}$$

$$2\eta + 1 - \text{TV} = \min_{h \in \mathcal{H}_p}\left(\text{err}(h; P_f) - \text{err}(h; Q_f)\right) + 1 \tag{49}$$

$$2\eta - \text{TV} = \min_{h \in \mathcal{H}_p} -\left(\text{err}(h; Q_f) - \text{err}(h; P_f)\right) \tag{50}$$

$$\text{TV} - 2\eta = \max_{h \in \mathcal{H}_p}\left(\text{err}(h; Q_f) - \text{err}(h; P_f)\right) = \xi \tag{51}$$

For Eq. 9, we use Eq. 48 and the above result to get the following

$$\epsilon_q = \max_{h \in \mathcal{H}_p} \text{err}(h; Q_f) = 1 - \min_{h \in \mathcal{H}_p} \text{err}(h; Q_{1-f}) \geq \text{TV} - 2\eta = \xi \tag{52}$$

We can write an inequality for errors similar to triangle inequality as follows

$$\text{err}(h; P_f) \leq \text{err}(h; P_g) + \text{err}(g; P_f) \tag{53}$$

$$= \text{err}(h; P_g) + \text{err}(f; P_g) = \text{err}(h; P_g) + \epsilon_f \tag{54}$$

$$\epsilon_p = \max_{h \in \mathcal{H}_p} \text{err}(h; P_f) \leq \max_{h \in \mathcal{H}_p} \text{err}(h; P_g) + \epsilon_f = 2\epsilon_f \tag{55}$$

The last equality follows from the definition of $\mathcal{H}_p$. Thus we get

$$\epsilon_q - \epsilon_p \geq \xi - 2\epsilon_f \tag{56}$$

By definition it follows that $\xi \geq \epsilon_q - \epsilon_p$ $\qquad\square$

**Proposition A.8.** *For $\beta > 0$, when the deteriorating shift occurs, for a chosen significance level of $\alpha$, the TPR of Algo. 2 is at least $(1 - \beta)\left(1 - \mathcal{O}\left(\exp\left(-n\epsilon_0^2 + d\right)\right)\right)$ if*

$$m \in \mathcal{O}\left(\left(\frac{1 + \sqrt{\delta}}{\xi - 2\epsilon_f}\right)^2 \left(d_p + \ln\frac{1}{\beta}\right)\right) \tag{57}$$

*and $\epsilon_q - \epsilon_p > 0$, where $\delta = \frac{d_p + \ln\frac{1}{\alpha}}{d_p + \ln\frac{1}{\beta}}$*

*Proof.* Similar to the proof of Theorem. 4.2, we derive the statistical power (TPR) of the test as follows. Let $\mu$ be the disagreement at $1 - \alpha$ percentile of $\Phi$

$$\text{TPR} = 1 - \Pr\left[\widehat{\text{err}}(h; Q_f) \leq \mu\right] \tag{58}$$

$$= 1 - \Pr\left[\{\{h \notin \mathcal{H}_p\} \wedge \{\widehat{\text{err}}(h; Q_f) \leq \mu\}\} \vee \{\{h \in \mathcal{H}_p\} \wedge \{\widehat{\text{err}}(h; Q_f) \leq \mu\}\}\right] \tag{59}$$

$$\geq 1 - \Pr\left[\{h \notin \mathcal{H}_p\} \vee \{\{h \in \mathcal{H}_p\} \wedge \{\widehat{\text{err}}(h; Q_f) \leq \mu\}\}\right] \tag{60}$$

$$\geq 1 - \Pr\left[h \notin \mathcal{H}_p\right] - \Pr\left[\{\widehat{\text{err}}(h; Q_f) \leq \mu\} \mid \{h \in \mathcal{H}_p\}\right] \Pr\left[h \in \mathcal{H}_p\right] \tag{61}$$

$$= (1 - \beta)\Pr\left[h \in \mathcal{H}_p\right] \tag{62}$$

$$\text{TPR} \in (1 - \beta)\left(1 - \mathcal{O}\left(\exp\left(-n\epsilon_0^2 + d\right)\right)\right) \tag{63}$$

where last equation comes from A.3 and $\beta := \Pr\left[\{\widehat{\text{err}}(h; Q_f) \leq \mu\} \mid \{h \in \mathcal{H}_p\}\right]$

Next, we derive the sample complexity $m$ in terms of $\beta$. We show that there exists a $\mu^*$ such that both A.2 (for $P$ and $\alpha$) and A.6 (for $Q$ and $\beta$) hold.

$$\epsilon_p < \mu < \epsilon_q \tag{64}$$

This implies some $\mu$ exists if $\epsilon_q - \epsilon_p > 0$

We find optimal $\mu^*$ such that the maximum of $m$ in Eq. 14 and Eq. 33 is minimized.

$$\left(\frac{\mu - \epsilon_p}{\epsilon_q - \mu}\right)^2 = \frac{d_p + \ln \frac{1}{\alpha}}{d_p + \ln \frac{1}{\beta}} := \delta \tag{65}$$

$$\mu^* = \frac{\epsilon_p + \sqrt{\delta}\epsilon_q}{1 + \sqrt{\delta}} \tag{66}$$

Plugging this $\mu^*$ in Eq. 33 gives

$$m \in \mathcal{O}\left(\frac{d + \ln \frac{1}{\beta}}{(\epsilon_q - \epsilon_p)^2}\left(1 + \sqrt{\frac{d + \ln \frac{1}{\alpha}}{d + \ln \frac{1}{\beta}}}\right)^2\right) \tag{67}$$

Use Eq. 9 to get the result. $\qquad\square$

$\xi$ in the denominator indicates that as the shift becomes more deteriorating, it is easier (fewer samples $m$) to monitor, indicating the effectiveness of the D-PDDM algorithm. Also, having a high-quality base classifier (low $\epsilon_f$) is better for D-PDDM which was also seen in Eq. 9 where low $\epsilon_f$ makes the algorithm more faithful. Note that $m$ depends on $d_p$ which can be much less than $d$ which $n$ depends on, suggesting that monitoring may be effective in few-shot settings. The dependence on $n$ is due to the requirement of satisfaction of condition 1 in Def. 2. In the optimization problems in Algo. 1, the empirical constraint is satisfied but the population constraint will be satisfied either for larger $\epsilon_0$ or for sufficiently large $n$ as seen in the theorem.

Next, we move to the regime where deteriorating shift occurs but $\epsilon_q - \epsilon_q \leq 0$. As a negative result, the following theorem states that in such cases the statistical power of the test is low.

**Theorem A.9.** *When deteriorating shift occurs and $\epsilon_q \leq \epsilon_p$, for a chosen significance level of $\alpha$, the statistical power of the test in Alg. 2 is $\mathcal{O}(\alpha)$.*

*Proof.* From the proof of Theorem. 4.4 we have

$$\text{TPR} \geq (1 - \beta)\left(1 - \mathcal{O}\left(\exp\left(-n\epsilon_0^2 + d\right)\right)\right) \tag{68}$$

$$\beta := \Pr\left[\{\widehat{\text{err}}(h; \boldsymbol{Q}_f) \leq \mu\} \mid \{h \in \mathcal{H}_p\}\right] \tag{69}$$

Using A.2 on $\boldsymbol{P}$ and $\alpha$ we get

$$\alpha \in \mathcal{O}(\exp\left(-n(\mu - \epsilon_p)^2 + d_p\right)) \tag{70}$$

Using A.2 on $\boldsymbol{Q}$ we get

$$\Pr\left(\{\widehat{\text{err}}(h; \boldsymbol{Q}_f) \geq \mu\} \mid \{h \in \mathcal{H}_p\}\right) \in \mathcal{O}\left(\exp\left(-n(\mu - \epsilon_q)^2 + d_p\right)\right) \tag{71}$$

$$1 - \beta \in \mathcal{O}\left(\exp\left(-n(\mu - \epsilon_q)^2 + d_p\right)\right) \tag{72}$$

$$1 - \beta \in \mathcal{O}\left(\alpha\right) \tag{73}$$

The last equation follows since we are dealing with the case where $\epsilon_p \geq \epsilon_q$. Thus, the TPR is $\mathcal{O}(\alpha)$ irrespective of the magnitude of $n$, as desired. $\qquad\square$

# B EXPERIMENTAL SETUP AND ADDITIONAL DETAILS

## B.1 THE BAYESIAN PERSPECTIVE

Crucial to the effective functioning of D-PDDM is the signal coming from the training set. It is this signal with which disagreement rates are computed in both D-PDDM pre-training and D-PDDM testing, as it serves as a "grounding" for the subsequent disagreement optimization in that one forces disagreement on a held-out validation set of choice which may or may not be in-distribution while constraining oneself to respect the training dataset. D-PDDM effectively translates this signal from the training set into a constrained hypothesis space $\mathcal{H}_p$ which gets outputted upon the completion of pre-training, thus avoiding the need to store the training dataset.

One can view the hypothesis space of a parametrized family as the parameter space themselves, though this is not yet enough. Even when one is able to represent $\mathcal{H}_p$, it then becomes a question of representing the restriction to this parameter space, i.e. fence the set of parameters such that models in this set perform well on the training dataset. To this, we propose a soft style of fencing by viewing the training of the base model from the Bayesian perspective. Specifically, in addition to optimizing the parameters of $f$, one also optimizes the posterior belief over the parameter space conditional on having observed the training data. Thus, lines 5 in Algorithm 1 and 2 of Algorithm 2 can be roughly approximated by sampling from the posterior.

**Maximum disagreement rate posterior sampling.** Let $\mathcal{H}$ be parametrized by $\mathcal{W}$ with some prior belief $P(w)$, denote the training dataset by $\mathcal{D}^n$. Upon training from $\mathcal{D}^n$, we update our belief over the weights of the model parameters via the posterior distribution $P(w|\mathcal{D}^n)$. For the pre-training step, let $\mathcal{D}^m$ be the in-distribution sample to disagree on at some round $t \leq T$. In order to approximate a disagreement rate conditional on our posterior, we sample weights $\{\tilde{w}_i\}_{i=1:K} \sim P(w|\mathcal{D}^n)$. Then, for each tentative weight $\tilde{w}_i$, we compute the tentative disagreement rate $\tilde{\phi}_i = \widehat{\text{err}}(h(\cdot; \tilde{w}_i); \mathcal{D}^m)$ on $\mathcal{D}^m$. Finally, $\max_{i=1:K} \tilde{\phi}_i$ is appended to $\Phi$. The exact same sampling procedure is used in Algorithm 2 in order to compute one disagreement rate $\phi_Q$ from $\mathcal{D}^m \sim Q$ where $P \overset{?}{=} Q$, where the algorithm identifies whether $\phi_Q$ lies beyond the $(1 - \alpha)$-quantile of $\Phi$. Effectively, although we do not solve exactly for the maximum disagreement rate achievable in the restricted family, we trade off this hard constraint requiring a potentially complicated optimization mechanism with an approximate lower-bound maximum sampling scheme by taking the empirical maximum disagreement rate from $K$ different posterior weight samples. It is also important to emphasize that this procedure implicitly relaxes the adherence on candidate auxiliary functions $h$ with weights $w$ having to be at least as good as $f$ on $\mathcal{D}^n$. Even with a concentrated posterior, the support of $P(w|\mathcal{D}^n)$ would still cover $\mathcal{W}$ and although unlikely, there is a possibility of sampling weights $w$ for which $h(\cdot; w)$ does not achieve the theoretical desired accuracy $\epsilon$. The result is, however, a very efficient sampling scheme where 1. we rely on the concentration of $P(w|\mathcal{D}^n)$ so that sampling extremely bad weights occurs rarely, and 2. we rely on a large $K$ in order to lower-bound the true maximum disagreement rate.

It is also worth mentioning that sampling from $P(w|\mathcal{D}^n)$ can be done via Markov Chain Monte Carlo, Langevin dynamics, variational inference, etc... Our implementation uses variational inference as it was found to be the fastest, though results may vary.

**Large models and very large models.** Though Algorithms 1 and 2 are largely theoretical, when viewed from the Bayesian perspective, the approximate optimization becomes tangible. When $\mathcal{H}$ is the family of neural networks on tabular features, the implementation effectively becomes Bayesian neural networks (Bishop, 1997). For neural networks accommodating various structural inductive biases on the data such as convolutional nets, recurrent nets, and potentially transformer architectures, per Harrison et al. (2024) it is possible to trade-off little to no performance in exchange for efficiency by modeling the distribution of parameters only on the last classification layer, thereby avoiding most of the variational inference on the feature extraction layers.

**Training dataset compression.** One may view the information content provided by the training set as being condensed into the constrained hypothesis space. We argue that for deep models trained on datasets of tens or hundreds of millions of samples, the storage and computation of disagreement rates represents a bottleneck in the efficient monitoring of algorithms. In training the models to

disagree, not only is the practitioner performing forward passes of the entire training set but also backpropagating the disagreement objective in order to finetune a version of the model from which a maximal disagreement rate is computed and appended to $\Phi$. Even when one entirely pretrains on more powerful HPC clusters, one is still left to run the D-PDDM test on a local machine monitoring the deployment of $f$. When considering monitoring language models or multimodal models in high-stakes environments for instance, both the training data storage and the backpropagation becomes a challenge.

The literature on coresets (Mirzasoleiman et al., 2020; Bachem et al., 2017; Karnin & Liberty, 2019; Feldman, 2020) provides a candidate solution to this problem. By sampling a small subset consisting of the most representative samples of the training set, one effectively compresses the information of the training set into rough representatives of the different classes. Similarly, the literature on prototype learning (Snell et al., 2017; Biehl et al., 2016; Xu et al., 2020; Deng et al., 2021) provide a similar style of information compression. In particular, prototypical ensembles may be employed where each ensemble member captures specific particularities of the training set, faithfully compressing not only the "average" but also the irregularities of the data manifold.

It could be worthwhile to consider $\mathcal{H}_p$ as feature representations of the training set instead. The entire training set is compressed via its feature representation which may be trained from supervision or in a semi-supervised fashion. This style of compression is conditional on having learned general representation of the dataset, the latter may be transfer learned from pretrained weights of similar tasks. In all of the aforementioned cases, one is still left to backpropagate during both pre-training as well as D-PDDM test. The Bayesian perspective avoids this entirely by offloading its computation into the approximate sampling scheme which can be done efficiently.

### B.2 SYNTHETIC DATA GENERATION

For all experiments, let $d$ be the number of features. To generate a $\boldsymbol{P}_g$-distributed dataset centered at $\boldsymbol{\mu} \in \mathbb{R}^{d-1}$ with isotropic variance $\sigma_1^2$, $(d-1)$-dimensional samples are generated from a Gaussian distribution with mean $\boldsymbol{\mu}$ and covariance $\sigma_1^2 \cdot \boldsymbol{I}_{d-1}$. For $n$ $(d-1)$-dimensional samples $\{\boldsymbol{x}^{(i)}\}_{i=1}^n$ with $\boldsymbol{x}^{(i)} = (x_1^{(i)}, \ldots, x_{d-1}^{(i)})$, we compute their $d^{th}$ features according to

$$x_d^{(i)} = \sum_{i=1}^{d-1} \sin(x_i^{(i)}) + \theta^{(i)} + \text{sgn}(\theta^{(i)}) \cdot \Delta$$

with $\theta^{(i)} \sim \mathcal{N}(0, \sigma_2^2)$, $\Delta \geq 0$ is a gap parameter, and $\boldsymbol{x}$ is then assigned the label $\text{sgn}(\theta^{(i)})$. Effectively our samples lie above and below a sinusoidal hypersurface decision boundary and are concentrated near $(\boldsymbol{\mu}, \sum \sin(\mu_i))$, with $\Delta$ controlling the minimum orthogonal distance from our samples to the decision boundary. In all experiments, we choose $\sigma_1^2 = \sigma_2^2 = 1$, and $\boldsymbol{\mu} = \pi \cdot \boldsymbol{I}_{d-1}$. For all experiments, the true positive rate (TPR) is reported at level $\alpha = 0.05$. In deteriorating shifts, we set $\Delta = 0$ while in non-deteriorating shifts, we set $\Delta$ to vary smoothly.

### B.3 INDUCING SHIFTS

**Deteriorating shift.** Deteriorating shift is induced by shifting $\boldsymbol{\mu}$ along the $(d-1)$-dimensional 1-vector by a factor of $\zeta$ which we smoothly control. From the data generation process, the first $(d-1)$ dimensional covariates are shifted and consequentially, the $d$-th covariate as well. We denote this shifted distribution by $\boldsymbol{Q}_x$. Effectively, this generates data centered near a different region of the decision boundary that the base classifier would not have seen and is unlikely to have been able to generalize to given the $\boldsymbol{P}_g$-distributed training data. In fact, as shown by **Figure 5**, we remark that it is highly unlikely that base classifiers could generalize the periodic property of the decision boundary given the low variance of the $x_1$ coordinate unless the hypothesis class exclusively contains periodic functions. Therefore, performance drop is expected for any base classifier that learns from the $\boldsymbol{P}_g$-distributed training samples.

**Non-deteriorating shift.** Non-deteriorating shift for the test distribution $\boldsymbol{Q}_x$ is induced by evenly mixing samples from $\boldsymbol{P}_g$ generated using $\Delta = 0$ with samples generated using $\Delta > 0$. This has the effect of stretching the centroids of the positively and negatively labeled points away from the

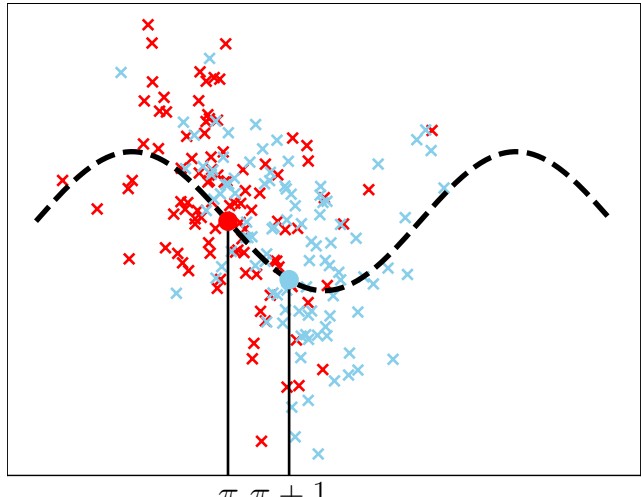

**Figure 7:** Data generation visualization in 2-dimensions. Samples generated from $\boldsymbol{P}_x$ are colored red, while samples generated from $\boldsymbol{Q}_x$ are colored blue. Samples above the sine curve are positively labeled, while those underneath are negatively labeled. The means of the first coordinates of distributions $\boldsymbol{P}_x$ and $\boldsymbol{Q}_x$ are labeled and colored accordingly. In practice, we analogously slide the mean of $\boldsymbol{Q}_x$ progressively to the right by $\zeta$ to induce deteriorating shift. The induction of non-deteriorating shift has an effect of stretching the distributions along the $y$-axis in the image as a function of the gap parameter $\Delta$.

decision boundary along the $d^{th}$ coordinate axis. A base classifier trained on $\boldsymbol{P}_g$ should perform similarly on $\boldsymbol{Q}_x$. We should expect a good PDD monitoring algorithm to enjoy low false positive rates under under this setup.

## B.4   ADDITIONAL SYNTHETIC RESULTS ON DETERIORATING SHIFT

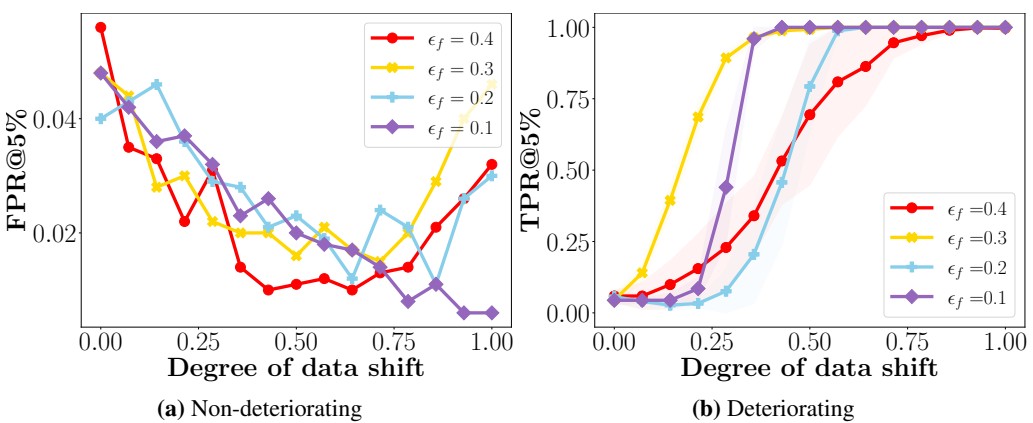

(a) Non-deteriorating

(b) Deteriorating

**Figure 8:** Analysis. **(a)** Across multiple base classifiers of differing qualities, the algorithm enjoys acceptably low FPRs when exposed to non-deteriorating shifts. **(b)** Base classifiers of differing qualities are able to effectively detect deteriorating shift and achieve high TPRs.

**Deteriorating shift: effectiveness of the D-PDDM.** We vary the amount of data shift in the x-axis for differing qualities ($\epsilon_f$) of the base classifier in Fig. 8 (b). For each base classifier, as the shift increases the test is better able to detect the deterioration. Note that deterioration in the shifts can be written as $\xi = \mathrm{TV} - 2\eta$ (from 5). The x-axis quantifies the amount of shift. However, the deterioration also depends on $\eta$ (see 6) which is different for each base classifier (even for fixed shift) due to differing $\epsilon_f$. Hence, for a specific data shift, the deterioration can differ across base classifiers. We see the

effect of this in 8 (b). In general, from 4.4, the trend is that a better base classifier (lower $\epsilon_f$) has higher TPR. For a particular shift, due to the difference in $\eta$ deterioration ($\xi$) could be different resulting in some deviation from the general trend. This highlights how 4.4 captures several of the subtle complexities of D-PDD.

## B.5 BASELINES DETAILS

We compare our disagreement-based hypothesis test algorithm against several other methods from the literature that either detect distribution changes or can be converted into a PDD monitoring protocol. Let $\boldsymbol{X} = \{\boldsymbol{x}^{(i)}\}_{i=1}^n$ from $\boldsymbol{P}_x$ and $\boldsymbol{Y} = \{\boldsymbol{y}^{(i)}\}_{i=1}^m$ from $\boldsymbol{Q}_x$ be given. These algorithms seek to accept or reject the hypothesis that $\boldsymbol{P}_x = \boldsymbol{Q}_x$ in distribution.

1. Deep Kernel MMD (Liu et al. (2020)) The algorithm first learns a deep kernel by optimizing a criterion which yields the most powerful hypothesis test. With this learned kernel, permutation tests are run multiple times in order to determine a true positive rate for the algorithm. We interface the authors' original source code with our repository and recycle their training procedures. Theirs can be found at https://github.com/fengliu90/DK-for-TST.

2. H-Divergence (Zhao et al. (2022)) The algorithm fits Gaussian kernel density estimates for $\boldsymbol{P}_x, \boldsymbol{Q}_x$, and their uniform mixture $(\boldsymbol{P}_x + \boldsymbol{Q}_x)/2$. Then, permutation tests are performed using the test statistic $H_\ell((\boldsymbol{P}_x + \boldsymbol{Q}_x)/2) - \min\{H_\ell(\boldsymbol{Q}_x), H_\ell(\boldsymbol{P}_x)\}$ where $H_\ell$ is the H-entropy with $\ell(x, a)$ the negative log likelihood of $x$ under distribution $a$ estimated by the Gaussian kernel density, in order to determine a true positive rate for the algorithm. This test statistic is an empirical estimate of the H-Min divergence. The choice of the particular H-divergence is a hyperparameter and is problem dependent, as well as the choice for how to generatively model the data distributions. The original paper further experimented with fitting Gaussian distributions as well as variational autoencoders (VAEs), both of which are not explored here. We interface the authors' original source code with our repository. Theirs can be found at https://github.com/a7b23/H-Divergence/tree/main.

3. $f$-Divergence (Acuna et al. (2021)) $f$-divergence generalizes several notions of distances between probability distributions commonly used in machine learning. In this paper, we convert the Kullback-Leibler (KL) and the Jensen-Shannon (JS) divergences, particular cases of $f$-divergences, into permutation tests. More specifically, we first fit Gaussians on samples coming from $\boldsymbol{P}_x$ and $\boldsymbol{Q}_x$ using maximum likelihood. In the case of KL-divergence, the empirical KL-divergence is computed between the fitted Gaussians whereas for the JS-divergence, we fit an additional Gaussian on the mixture distribution $\boldsymbol{M}$ and leverage the identity:

$$\mathrm{JS}(\boldsymbol{P}_x || \boldsymbol{Q}_x) = \frac{1}{2}(\mathrm{KL}(\boldsymbol{P}_x || \boldsymbol{M}) + \mathrm{KL}(\boldsymbol{Q}_x || \boldsymbol{M}))$$

We run permutation tests by permuting the samples in the union $(X \sim \boldsymbol{P}_x^n) \cup (Y \sim \boldsymbol{Q}_x^m)$. It is worth noting that as with H-divergence, more elaborate generative models could be fitted onto samples $X$ and $Y$, which we do not explore in this work.

4. Black Box Shift Detection (BBSD) (Lipton et al., 2018) involves estimating the changes in the distribution of target labels $p(y)$ between training and test data while assuming that the conditional distribution of features given labels $p(x|y)$ remains constant. This is achieved by using a black box model's confusion matrix to identify discrepancies in the marginal label probabilities between the training and test distributions, allowing detection and correction of the shift.

5. Relative Mahalanobis Distance (RMD) (Ren et al., 2021) RMD modifies the traditional Mahalanobis Distance (MD) for out-of-distribution (OOD) detection by accounting for the influence of non-discriminative features. It subtracts the MD of a test sample to a background class-independent Gaussian from the MD to each class-specific Gaussian, effectively isolating discriminative features and improving OOD detection, especially for near-OOD tasks. We test for shift by performing a KS test directly on the distribution of the RMD confidence scored computed on $\boldsymbol{Q}_x$ and $\boldsymbol{P}_x$.

Importantly, most baselines perform some comparison using distances in covariate space. Though this may be effective, they are inevitably susceptible to false positives when the

shift is non-deteriorating. To the best of our knowledge, our method is the first deteriorating shift detection method which resists flagging non-deteriorating shifts due to leveraging the disagreement statistics which help the model mitigate FPRs down the line, as the auxiliary models would not disagree any better than the base model out-of-distribution when the distribution does not result in model deterioration.

## B.6 THE GEMINI DATASET

**GEMINI Study and Preprocessing.** The General Medicine Inpatient Initiative (GEMINI) study is a retrospective cohort study of adult patients and their clinical and administrative data (Verma et al., 2021). This analysis used data from over 200,000 patients from the GEMINI Database, spanning 7 different hospitals that participated in the GEMINI Study. Each patients information is processed into 900 features including but not limited to: (i) laboratory results and vital results collected up to 48-hours after admission, split into 6 hour intervals, (ii) patient demographic information: age, sex etc, (iii) Patient diagnosis using ICD-10-CA codes. Missing feature values are imputed based on simple averaging. The predictive task related to this data is to predict 14-day mortality for patients based on these collected features.

**Data Splitting and Shift.** Based on this pre-processed data, 2 shifts are analysed: (i) temporal shift, and (ii) age-group shift. The temporal shift analysis splits data into half-years - 2018H1, 2019H2, etc. The baseline model uses 2017H1 and prior data for training, and 2017H2 for validation; Tab. 3 shows patient statistics for this split. It is subsequently tested on unseen in distribution data and later splits. The different age groups are created by splitting the data into 5 equally sized groups based on ages of patients: (1) 18-52, (2) 52-66, (3) 66-72, (4) 76 - 85, (5) 85+; Tab. 4 shows patient statistics for this split. The reported analysis trains a baseline model on group 1 (18-52) and then tests on test-sets that contain some portion of data from the 5th group (85+) and the remaining as unseen in distribution data. The portions [0.0, 0.2, 0.4, 0.6, 0.8, 1.0] represent what percentage of the test set is ood (from group 5), whilst the remaining amount is iid (from group 1). For example a ratio of 0.2 means 20% of the test data is from group 5(ood) whilst 80% is from group 1(iid). We chose to experiment on such portions instead of just subsequent age groups as this process better displays the True Positive Detection Rate of our method as well as baselines w.r.t degree of shift / perfromance deterioration.

**Model and Method Hyper-Parameters.** The base models used were neural networks with hidden layers [128, 64, 32, 16] and were trained to predict 14 day mortality on patients given the aforementioned co-variates. The hyper parameters of the model and method were fixed to be the same for both the age and temporal contexts to ensure fairness of results, thus showing that at a fixed sensitivity, the method is able to differentiate between deteriorating and non-deteriorating shifts. Our recommendation is to use a control test-set with a known deterioration in order to tune hyper parameters to desired sensitivity to distribution shift.

| Year | Patient Count | Label Ratio |
|------|---------------|-------------|
| Pre-2017 | 72316 | 3.99% |
| 2017H2 | 17208 | 3.60 % |
| 2018H1 | 18233 | 4.15% |
| 2018H2 | 18469 | 3.83% |
| 2019H1 | 19041 | 3.50% |
| 2019H2 | 18601 | 3.49% |
| 2020H1 | 15575 | 4.50% |
| 2020H2 | 11155 | 3.48% |
| 2021H1 | 10625 | 3.46% |
| 2021H2 | 7396 | 2.95% |

**Table 3:** Temporal Split Data Summary

| Age | Patient Count | Label Ratio |
|------|---------------|-------------|
| 18-52 | 33220 | 0.82 % |
| 52-66 | 33146 | 2.36 % |
| 66-72 | 31048 | 3.36 % |
| 76 - 85 | 34055 | 4.77 % |
| 85+ | 32399 | 7.82 % |

**Table 4:** Age Split Data Summary

## B.7 STATEMENT ON THE USAGE OF COMPUTING RESOURCES

All experiments were run on High Performance Computing (HPC) clusters. For our algorithm as well as some of the baselines, neural networks used as function approximators are implemented in

PyTorch and trained on GPU-enabled nodes. The requested memory for all compute jobs was 16G each, but we believe the jobs are able to run with much less memory. Time of execution was not documented throughout the process.