# OpenReview forum: "Provable Post-Deployment Deterioration Monitoring"
_ICLR.cc/2025/Conference — Submitted to ICLR 2025_

### Official Review · Reviewer_kjuu · 2024-10-31

**Soundness:** 3
**Presentation:** 2
**Contribution:** 3
**Rating:** 6
**Confidence:** 2

**Summary:**

This paper proposes an unsupervised algorithm, D-PDDM, based on model disagreement for monitoring performance deterioration of machine learning models after deployment. The method detects performance degradation by measuring the prediction discrepancies between a well-performing auxiliary model and the deployed model in an unlabeled deployment environment, without requiring training data during deployment, thereby ensuring privacy and scalability. Theoretical analysis demonstrates that the algorithm maintains a low false positive rate under non-deteriorating data shifts and effectively detects deteriorating shifts. Experimental results show that it outperforms existing methods across various datasets.

**Strengths:**

1.	The approach of leveraging model disagreement for unsupervised PDD is a solution to the problem of monitoring performance without access to labels. It innovatively combines existing ideas from distribution shift literature and disagreement frameworks.
2.	The paper provides a solid theoretical foundation for the proposed method. The mathematical analysis is detailed, covering both deteriorating and non-deteriorating shifts. Additionally, the authors anticipate potential failure scenarios and propose strategies to mitigate them.
3.	The paper is well-structured, with clear definitions and thorough explanations of the proposed method. The inclusion of empirical results across diverse datasets enhances the comprehensibility of the theoretical guarantees.
4.	The work addresses a critical gap in real-world machine learning deployment, where acquiring labels is often impractical or costly. The proposed method's ability to operate without training data makes it particularly relevant for privacy-sensitive domains like healthcare.

**Weaknesses:**

1.	While the paper emphasizes not requiring training data during deployment, the approach's reliance on pre-trained auxiliary models raises questions about generalizability. If the auxiliary model does not generalize well to the deployment environment, D-PDDM's effectiveness could be compromised.
2.	The method's reliance on hypothesis spaces and sampling might face challenges in complex, high-dimensional settings. The scalability claims, though theoretically sound, might need further empirical validation, especially in more intricate real-world scenarios.
3.	Assumptions in Theoretical Analysis: The theoretical guarantees hinge on assumptions such as small total variation distances between distributions. These assumptions might not always hold in practice, especially in environments with drastic changes, potentially limiting the method's applicability.

**Questions:**

1.	How does the choice of auxiliary model affect the performance of D-PDDM? Would certain types of models or training procedures make the method more robust?
2.	Can the authors provide more insights or empirical results on how the algorithm scales when dealing with complex, high-dimensional data? Are there particular datasets where D-PDDM might struggle?
3.	The threshold for determining significant disagreement is crucial for the performance of D-PDDM. How sensitive is the method to this threshold, and are there automated ways to set it optimally?
4.	The paper suggests robustness to non-deteriorating shifts, but how does the method handle abrupt, substantial changes in the data distribution, which might not be gradual? Would the disagreement framework still provide reliable monitoring in such scenarios?

---

> ### Author Response · Authors · 2024-11-22
> **Response to Reviewer kjuu (1/2)**
>
> We would like to thank the reviewer for your helpful and detailed comments. We greatly appreciate the reviewer’s time and effort, and are pleased to hear that the reviewer finds that our approach “innovatively combines existing ideas from distribution shift literature and disagreement frameworks”, that our work “provides a solid theoretical foundation”, and that our work “addresses a critical gap in real-world machine learning deployment”. In the following we hope to address some concerns raised in their review.
>
> ### Weaknesses concerns
>
> >While the paper emphasizes not requiring training data during deployment, the approach's reliance on pre-trained auxiliary models raises questions about generalizability. If the auxiliary model does not generalize well to the deployment environment, D-PDDM's effectiveness could be compromised.
>
> We choose $\mathcal{H}_p$ to be rich enough, e.g. the class of neural networks. In particular, the general class $\mathcal{H}$ is also where $f$ comes from. Generalization to the deployment environment is not a consideration for the functionality of our method given that the deployment distribution does not contain ground truth labels. The requirement is that the general class $\mathcal{H}$ (and thus the restricted subset $\mathcal{H}_p$ ) is expressive enough to disagree with $f$, as the method relies purely on the statistics of the in-distribution disagreement rates against the deployment distribution disagreement rate.
>
> >The method's reliance on hypothesis spaces and sampling might face challenges in complex, high-dimensional settings. The scalability claims, though theoretically sound, might need further empirical validation, especially in more intricate real-world scenarios.
>
> Thank you for pointing this out! We agree that our current work is heavily focused on novel theoretical analysis. We believe that in practice, the hypothesis space can be translated to complex scenarios. For example, the (deep) Bayesian framework could be implemented. In this way, the output of Algorithm 1 is effectively a distribution of network parameters over which one searches to maximize disagreement rates.
>
> Though we do not provide studies on the effectiveness of D-PDDM on very large-scale architectures, we remark that the Bayesian perspective can once again be adopted, although not without some caveats. As per [1], it may suffice to represent the distribution of the last classification layer. This effectively trades off a bit of performance for significant efficiency with regards to space as storing variational information of the feature extraction stage can be entirely avoided.
>
> A more detailed and thorough discussion revolving around the Bayesian implementation of D-PDDM as well as technical considerations for its potential deployment in real-world scenarios will be supplied in the revised version of the Appendix as well as properly referred to in the main paper.
>
> > Assumptions in Theoretical Analysis: The theoretical guarantees hinge on assumptions such as small total variation distances between distributions. These assumptions might not always hold in practice, especially in environments with drastic changes, potentially limiting the method's applicability.
>
> Thank you for bringing this up. We would first like to highlight that the theoretical assumptions here are designed in order to render the analysis possible. In real-world scenarios, even when the assumptions are relaxed or not fully met, our method can be robust with respect to deteriorating and non deteriorating changes, as evidenced by the empirical results on the GEMINI clinical dataset. Our future work evaluates deployments of D-PDDM on a wide range of datasets of varying data modalities in order to assess its empirical capabilities beyond the guarantees provided by the theory.
>
> ### References
>
> [1] Harrison, James, John Willes, and Jasper Snoek. "Variational bayesian last layers." arXiv preprint arXiv:2404.11599 (2024). ICML 2024 (Spotlight)

---

> ### Author Response · Authors · 2024-11-22
> **Response to Reviewer kjuu (2/2)**
>
> ### Addressing questions
>
> >How does the choice of auxiliary model affect the performance of D-PDDM? Would certain types of models or training procedures make the method more robust?
>
> As mentioned in Section 3, there are a few ways to go about implementing D-PDDM in practice. We take the Bayesian approach for neural networks, thus translating the auxiliary class $\mathcal{H}_p$ into a posterior distribution over network parameters. Regarding the robustness of the method, in general we found that employing a rich class of parametric models over which gradient-based optimization can be run is sufficient for the robustness of the method. Importantly, as long as models in the hypothesis class can reasonably disagree in-distribution, the deteriorating changes can effectively be monitored. In contrast, when the hypothesis class $\mathcal{H}$ is inexpressive, one may find that the distribution of in-distribution disagreement rates collapse, and thereby testing deployment disagreement on its 95th quantile returns inconclusive results.
>
> >Can the authors provide more insights or empirical results on how the algorithm scales when dealing with complex, high-dimensional data? Are there particular datasets where D-PDDM might struggle?
>
> When dealing with high-dimensional data, one could still reasonably employ the D-PDDM framework for monitoring. As mentioned previously, a Bayesian approach of modeling the distribution of the last classification layer may be applied on a separately pretrained feature classification layer. In this way, we can address multiple structured data modalities while remaining consistent with our framework. We found, however, that without proper encoding of categorical features in a tabular dataset, D-PDDM tends to be more susceptible to false positives when the distribution shifts are non-deteriorating.
>
> >The threshold for determining significant disagreement is crucial for the performance of D-PDDM. How sensitive is the method to this threshold, and are there automated ways to set it optimally?
>
> The threshold for determining significant disagreement is set automatically to be the 95th quantile of the (empirical) distribution of in-distribution disagreement rates. Depending on the dataset as well as the expressivity of the model class $\mathcal{H}$ of choice and the disagreement training procedure, this distribution can look very different. The 95th percentile is selected as per the null hypothesis whern there is no distribution shift, the approximate probability of observing an unusually high disagreement rate is 5%, which is our desired false positive rate (FPR). The specific choice can be attuned to the preferences of the practitioner as a function of the predictive task they are solving.
>
> >The paper suggests robustness to non-deteriorating shifts, but how does the method handle abrupt, substantial changes in the data distribution, which might not be gradual? Would the disagreement framework still provide reliable monitoring in such scenarios?
>
> Thank you for pointing this out! Substantial changes in the data distribution may or may not be deteriorating. Both of these scenarios are captured and addressed by the theoretical results 4.2 and 4.4. Therefore, D-PDDM provides reliable monitoring in such scenarios.
>
> In practice, we found that large substantial changes that are deteriorating are easily flagged by D-PDDM. On rare occasions, small gradual changes that are deteriorating may be harder to flag due to an unlikely sampling of the deployment distribution. These are also addressed in full by the probabilistic guarantees provided by the sample complexity analyses in Section 4.
>
> ### Closing comments
>
> We thank the reviewer again for their valuable feedback on our work. We hope that our rebuttal addresses their questions and concerns. A revised copy of our submission and supplementary will be uploaded and notified via the global response thread in the coming days. We kindly ask the reviewer to consider a fresher evaluation of our paper if the reviewer is satisfied with our responses. We are also more than happy to answer any further questions that arise that might improve your assessment of the work.

---

> ### Comment · Reviewer_kjuu · 2024-11-26
> **Response**
>
> Thanks for authors' detailed responses. I think that they have adressed my concerns. I will remain my positive score. Good Luck!

---

### Official Review · Reviewer_gX35 · 2024-11-03

**Soundness:** 3
**Presentation:** 3
**Contribution:** 3
**Rating:** 5
**Confidence:** 4

**Summary:**

The authors explore the impact of data distribution shifts on model performance after models are deployed in new environments, focusing specifically on shifts that do not lead to performance degradation, thus avoiding unnecessary retraining. The paper proposes a monitoring algorithm called D-PDDM (Disagreement-based Post-Deployment Deterioration Monitoring) to detect performance deterioration (PDD) in the unlabeled deployment distribution. This algorithm is based on a model inconsistency framework: if an auxiliary model performs well on the training data but exhibits significant predictive inconsistency with the deployed model on the test data, performance deterioration is detected.

**Strengths:**

- The method proposed by the authors to address this problem is quite interesting; specifically, to avoid accessing the original dataset and to evaluate the performance of the trained model, an auxiliary model is used to "memorize" information about the original distribution and performance differences.
- The paper is well-organized, making it clear and easy to understand.
- The theory presented seems to be largely sound.

**Weaknesses:**

- I'm not sure if it's my mistake, but it seems the authors did not upload the code required to reproduce this work (the anonymous link does not seem to include the code).
- The experimental section lacks an introduction to the comparison methods, which may make it unclear for non-experts in this field whether the comparisons are reasonable or if they include the latest methods in this domain, as the proposed method only appears to be compared with several distribution divergence-based detection methods.
- The algorithm proposed in the paper may require substantial computational resources, especially during the pre-training phase. This could limit its practicality in resource-constrained environments, and the authors do not seem to mention an appropriate size for the sub-hypothesis space.

**Questions:**

- To address the issues mentioned in this paper, could continual learning or online learning serve as a final solution?
- For a large-scale model $f$, does the auxiliary model $h$ necessarily need to have the same structure as $f$?
- Is the proposed method highly dependent on the auxiliary model $h$'s ability to effectively fit the output of $f$?
- When the model's performance declines despite no significant distribution shift, could this be attributed to overfitting? Would this method still be effective in such cases?

---

> ### Author Response · Authors · 2024-11-22
> **Response to Reviewer gX35 (1/3)**
>
> We would like to thank the reviewer for their helpful and detailed comments. We greatly appreciate the reviewer’s time and effort, and are encouraged to hear that the reviewer finds our method’s leveraging of information from the training set “quite interesting”, that our work is “clear and easy to understand”, and that our theory is “largely sound”. In the following, we hope to address some concerns raised in their review.
>
> ### Weaknesses concerns
>
> >I'm not sure if it's my mistake, but it seems the authors did not upload the code required to reproduce this work (the anonymous link does not seem to include the code).
>
> Thank you for pointing this out. The code is uploaded to the appropriate repository. The hyperlink is included here for convenience: https://anonymous.4open.science/r/d_pddm-F966/
>
> >The experimental section lacks an introduction to the comparison methods, which may make it unclear for non-experts in this field whether the comparisons are reasonable or if they include the latest methods in this domain, as the proposed method only appears to be compared with several distribution divergence-based detection methods.
>
> Thank you for this suggestion. Indeed, our paper tackles a novel problem: monitoring a model’s performance for different downstream datasets. This requires two key properties: sensitivity in detecting performance deterioration and robustness in resisting false alarms when presented with non-deteriorating shifts. The current literature, including distribution-divergence-based methods, often implicitly assumes that all distribution shifts result in performance drops, which is clearly not always the case. This gap motivated our work, and to the best of our knowledge, we are the first to theoretically address the robustness required for non-deteriorative shifts. Thus, finding the best baselines in our particular context is challenging in that the extent to which algorithms in the related literature embodying different sets of assumptions could reasonably perform in the settings of interest to us is nuanced. Hence, we opted for comparisons with the most popular divergence-based methods.
>
> For completeness, the experiments comparing D-PDDM with a few other traditional out-of-distribution (OOD) detection baselines on synthetic as well as on the GEMINI clinical dataset are on the way, and we hope to append these results to the global rebuttal thread in the coming days.
>
> In the revised version of the manuscript, we will make the corresponding changes in Section 5 per your suggestions on including an introduction to the comparison methods:
> - Properly refer the synthetic data generation procedure to its corresponding section in the supplementary materials
> - Properly refer the Implementation & Baselines paragraph to its corresponding section in the supplementary materials
> - Append detailed descriptions of the ongoing baselines to Appendix B.4
> - A more detailed discussion on baseline selection will be provided as per the above.
>
> >The algorithm proposed in the paper may require substantial computational resources, especially during the pre-training phase. This could limit its practicality in resource-constrained environments, and the authors do not seem to mention an appropriate size for the sub-hypothesis space.

---

> ### Author Response · Authors · 2024-11-22
> **Response to Reviewer gX35 (2/3)**
>
> >The algorithm proposed in the paper may require substantial computational resources, especially during the pre-training phase. This could limit its practicality in resource-constrained environments, and the authors do not seem to mention an appropriate size for the sub-hypothesis space.
>
> Thank you for highlighting this concern. We aimed to provide the most general version of our algorithm in Section 3 to be consistent with the theoretical analysis in Section 4 and Appendix A. Since argmax-ing over the subset of the hypothesis Hp can be done in several ways, we propose running this optimization using a Bayesian framework, thereby optimizing Bayesian neural networks. Alternatively, BatchEnsemble or schemes involving the sampling of representative samples of the training set may be explored as part of our future work in progress.
>
> The Bayesian perspective is especially helpful in resource-constrained environments, as the storage of a subset of the hypothesis space regresses to storing a posterior distribution over the space of function approximators of choice as well as samples from it. When deep neural networks are employed, as [1] suggests it suffices to store distribution of the last layer, thereby completely avoiding storing Bayesian posteriors of the feature extraction layers. Therefore, although the desired subset of the hypothesis space may be complex, the added computational cost scales linearly in the complexity of the final classification layer.
>
> We will include details about the Bayesian implementation in the supplementary and refer accordingly in Section 3.
>
> Specifically, in the supplementary:
>
> - We will append details regarding the Bayesian perspective where we view the restricted hypothesis space Hp as a posterior distribution, effectively casting the pretraining algorithm (Algorithm 1) into training Bayesian neural networks and sampling from its posterior distribution.
>
> - Details and discussions regarding approximate sampling schemes as well as suggested best practices will be included.
>
> - Discussions regarding the translation of the theoretical D-PDDM algorithms into practice in resource-constrained environments via the Bayesian perspective and [1] will also be included.
>
> ### Addressing questions
>
> >To address the issues mentioned in this paper, could continual learning or online learning serve as a final solution?
>
> Thanks for your insightful thoughts! We agree that continual learning (CL) is one possible solution in the adaptation if the sequential downstreaming tasks are always deteriorating, necessitating either continual adaptation or an ever-changing policy for an artificial agent. One can view the continual monitoring of a machine learning model in deployment as a decision problem where at each timestep, the agent is to make a decision of whether to re-train the base model or not. Alternatively, one can also view the agent’s decision space as all possible changes to the model it can induce, thereby casting continual adaptation as a decision problem. We believe that future work in CL for monitoring could be very interesting given that one derives an appropriate reward shaping mechanism by which current state-of-the-art CL agents may learn effectively.
>
> >For a large-scale model f, does the auxiliary model h necessarily need to have the same structure as f?
>
> Thank you for this insightful question. In our theoretical analysis, $f$ and $h$ are within the same hypothesis space, i.e. the same structure. In addition, we also believe that the integrity of the algorithm may not be immediately compromised when the auxiliary space misspecifies $f$. For example, inspired by LIME [2],  $f$ is a large neural network and $\mathcal{H}$ is a linear class aimed at capturing local information. We can split the data distribution into the several sub-distributions, where we in turn compute the local disagreements between $f$ and $h \in \mathcal{H}_p$ . Then, global disagreement can still be reliably estimated as per [2]. Though this particular setup is beyond our theory, it may still be effective in practice.
>
> ### References
>
> [1] Harrison, James, John Willes, and Jasper Snoek. "Variational bayesian last layers." arXiv preprint arXiv:2404.11599 (2024). ICML 2024 (Spotlight)
>
> [2] Marco Tulio Ribeiro, Sameer Singh, Carlos Guestrin, “Why Should I Trust You?: Explaining the Predictions of Any Classifier” KDD 2016

---

> > ### Comment · Reviewer_gX35 · 2024-11-27
> >
> > Thank you very much for the authors' response, which has resolved most of my questions.
> >
> > However, after reviewing the feedback from other reviewers, I realize that the implementation details of the auxiliary model are very important, yet the paper does not elaborate on them sufficiently. The overhead introduced by the auxiliary model is the main cost of the proposed method, but the authors seem to assume that the auxiliary model and the original model are of the same origin. If this is the case, then when
> > $f$ is large, it will result in significant overhead in terms of storage and computation time. Therefore, I raised the question, "For a large-scale model $f$, does the auxiliary model $h$ necessarily need to have the same structure as $f$"? However, the authors consider this scenario beyond their theoretical scope and did not provide further results. As a result, I will maintain my current score.
> >
> > It is noteworthy that this discussion is crucial because whether a more lightweight component can achieve the same effect determines the practical significance of the proposed method.

---

> ### Author Response · Authors · 2024-11-22
> **Response to Reviewer gX35 (3/3)**
>
> >Is the proposed method highly dependent on the auxiliary model  h 's ability to effectively fit the output of f?
>
> We find that the proposed method is moreso dependent on the expressivity of the hypothesis class of choice rather than the ability to fit the (inversed) output of $f$. As an extreme example, when the hypothesis class contains two elements that identically output $1$ or $0$ for any covariates, the distribution of disagreement rates for in-distribution samples may collapse to $1$, thus results from a subsequent D-PDDM test on a deployment sample would be uninformative. As can be seen in our response to the reviewer’s prior question, D-PDDM may practically work even when $\mathcal{H}$ misspecifies the base classifier $f$, as long as $\mathcal{H}$ is flexible enough.
>
> >When the model's performance declines despite no significant distribution shift, could this be attributed to overfitting? Would this method still be effective in such cases?
>
> Thank you for this great remark. In this case, in-distribution performance degradation can be attributed to overfitting. Assuming we have the same training procedure for the base classifier $f$ and the auxiliary disagreement classifiers $h$ from a very rich class $\mathcal{H}$, if $f$ overfits on training data, we are likely to have the $h$’s overfit on the inverse outputs of $f$, thereby achieving extremely high disagreement rates. It is likely that on in-distribution deployment samples, the same procedure will observe similarly high disagreement rates, thereby rendering our test uninformative. As such, our method does not address the i.i.d. scenario coupled with a base model deterioration. The application of the disagreement framework for the i.i.d. scenario has been investigated in [1].
>
> ### Closing comments
>
> We thank the reviewer again for their valuable feedback on our work. We hope that our rebuttal addresses their questions and concerns. A revised copy of our submission and supplementary will be uploaded and notified via the global response thread in the coming days. We kindly ask the reviewer to consider a fresher evaluation of our paper if the reviewer is satisfied with our responses. We are also more than happy to answer any further questions that arise that might improve your assessment of the work.
>
> ### References
>
> [1] Jiang, Yiding, et al. "Assessing generalization of SGD via disagreement." arXiv preprint arXiv:2106.13799 (2021). ICLR 2022 (Spotlight)

---

### Official Review · Reviewer_CJWB · 2024-11-03

**Soundness:** 2
**Presentation:** 3
**Contribution:** 3
**Rating:** 6
**Confidence:** 3

**Summary:**

The accuracy of ML models predictions can deteriorate after deployment due to changes in the distribution of the feature distribution. The paper proposes that this deterioration can be detected indirectly without using labeled deployment data. It offers a method built of two steps: a) identify, in addition to the chosen model, a reference set of high-performing prediction models at training time; and b) check the maximal discrepancy, in terms of predictions on deployment data, between the reference models and the chosen model.
The paper provides a theorem to show conditions under which prediction disagreement with other good models is “equivalent” to predictions deteriorating compared to the training-set. It then proposes a method to calibrate a deterioration test from final samples, together with theoretical results regarding the rates by which the detection rates converge when deterioration occurs and when it does not. Experiments are shown comparing the FPR and TPR to other methods on a synthetic and two real datasets.

**Strengths:**

Significance: The problem of detecting changes in the performance of  prediction algorithms across environments is a fundamental challenge in machine learning, and algorithms that would detect such changes before observing the true labels are attractive because often obtaining such labels requres deliberate allocation of resources. The paper outlines an approach to tackling the problem using "critic" models as surrogates for the ground truth, and develops some theory for when we would expect the approach to be successful. At current state, the algorithm is more a proof of concept than ready-made (some considerations such as choice of critic models, performance when prediction error is non-negligible, are not fully discussed or developed)

Originality: The disagreement approach suggested by the paper is somewhat new in the context discussed:
identifying a critic function that agrees on the training set but diverges at deployment.
I should note however that the results are quite similar in flavour to the ideas in Rosenfeld and Garg 2023, using the relations (and similar mathematical arguments) found in that paper - an upper bound on prediction after distribution shift - but in reverse.

[Also - If indeed the authors agree that the fundamental results used are based closely on the relations described in RG2023, perhaps they should have made a more explicit attribution].

Quality: The paper puts together several ideas in an easy to follow way. The theoretical explanation are illustrative. I later point areas in the theoretical results which may have small issues (or perhaps are unclear to me).

Clarity: The paper is well written and easy to follow.
I think some of the use of the statistical naming conventions is half-hearted, adding to the confusion.
As the main example, the terms "significance" (and "alpha") are not used explicitly in a statistical
context, and so I was left to wonder until quite late whether indeed the FPR is kept at alpha etc. (If I am not mistaken it is, but
I think being explicit would have helped).

**Weaknesses:**

Some of the discussion of weaknesses is already there in the description of the strengths:
- The algorithm is more of a sketch, for example in how to choose Hp.
- The paper has novelty, but the results are not a far extension of RG2023.
- It is not clear if it is useful if we don't have a very accurate model.

Other than that, I have some questions detailed in the next section:

**Questions:**

1. Isn’t it too good to be to be true to assume that g the ground truth function is in H?
At first seemed to me a techicality, but when I look at the proof fo Lemma 2.1 (in the supplementary), PDD->DPDD,
it hedges on g being the that function. But is g achievable in practice? and if not, what are implications?

2. I don't quite understand what is the probabilitic statement hiding here
"With high probability, PDD is equivalent to D-PDD".
This is I think a similar probability  model as here
"As this equivalence happens in probability, Def. 2 allows for certain false positive errors w.r.t. to Def. 1"
and here
"We need to demonstrate h = g with high probability" (in proof of the Equivalence condition, Lemma A.1).

But to me It seems that expressions such as "err(f, Qg)"  and "err(f,Pg)" are not supposed to be probabilistic statements,
because f and g are fixed functions and this is an expected statements.

---

> ### Author Response · Authors · 2024-11-22
> **Response to Reviewer CJWB (1/2)**
>
> We would like to thank the reviewer for your helpful and detailed comments. We greatly appreciate the reviewer’s time and effort, and are pleased to hear that the reviewer finds that our method addresses a “fundamental challenge in machine learning”, that our work is “well written and easy to follow”, and appreciate its originality. In the following we hope to address some concerns raised in their review.
>
> > As the main example, the terms "significance" (and "alpha") are not used explicitly in a statistical context, and so I was left to wonder until quite late whether indeed the FPR is kept at alpha etc. (If I am not mistaken it is, but I think being explicit would have helped).
>
> Thank you for pointing this out. In the revised version of the main paper, we will make sure to relate “significance” and “alpha” in describing FPR and TPR for non-deteriorating and deteriorating shifts respectively from the beginning of Section 4 as to avoid confusion.
>
> > The algorithm is more of a sketch, for example in how to choose Hp.
>
> Thank you for raising this important concern. Indeed, we aim to provide the most general version of our algorithm in Section 3 to be consistent with the theoretical analysis in Section 4 and the Appendix A. Since argmax-ing over the subset of the hypothesis Hp can be done in several ways, we propose running this optimization using a Bayesian framework, thereby optimizing Bayesian neural networks. Alternatively, BatchEnsemble or schemes that sample a small representative subset of the training set may be explored as part of our future work in progress. We will include details about the Bayesian implementation in the supplementary and refer accordingly in Section 3.
>
> Specifically, in the supplementary:
>
> - We will append details regarding the Bayesian perspective where we view the restricted hypothesis space Hp as a posterior distribution, effectively casting the pretraining algorithm (Algorithm 1) into training Bayesian neural networks and sampling from its posterior distribution.
> - Details and discussions regarding approximate sampling schemes as well as suggested best practices will be included.
>
> The inclusion of the detailed description as well as discussion around the Bayesian implementation will provide clarification on the potential practical impacts of the work. In particular, since we applied approximate sampling in order to compute disagreement statistics, the exact tradeoff with regards to the baseline of making the training set available in the D-PDDM test will be discussed. Along with this, the tradeoff of the storage of posterior samples of the Bayesian neural networks against storing the training dataset will be further elucidated.
>
> >The paper has novelty, but the results are not a far extension of RG2023.
>
> Thank you for these remarks. We agree that RG2023 [1] provides relevant context, and we would like to clarify the distinctions between our work and theirs. RG2023 primarily focuses on accurately estimating the test dataset error through domain adaptation theory. A key underlying assumption in RG2023 is that a performance drop occurs in the target domain, guiding their methodology. In contrast, our work addresses performance monitoring in a broader context where performance may either improve or degrade due to distribution shifts. Thus, our objective is twofold: (1) to achieve high sensitivity with respect to detecting performance drops (aligned with RG2023) and (2) to maintain robustness against non-deteriorative shifts, a novel contribution not addressed in RG2023. We will ensure that this distinction is more clearly discussed in the revised version of the paper in the Related Work section where more explicit attributions to RG2023 will be provided.
>
> ### References
>
> [1] Rosenfeld, Elan, and Saurabh Garg. "(Almost) Provable Error Bounds Under Distribution Shift via Disagreement Discrepancy." Advances in Neural Information Processing Systems 36 (2023): 28761-28784.

---

> > ### Comment · Reviewer_CJWB · 2024-11-26
> >
> > Thank you for your responses.
> > The answers have not increased my confidence in the current version of the paper,
> > which in I think would benefit from a revision (and I think you allude to some expected changes as well).
> > I will leave the score unchanged.
> >
> > " Since argmax-ing over the subset of the hypothesis Hp can be done in several ways, we propose running this optimization using a Bayesian framework, thereby optimizing Bayesian neural networks."
> >
> > That's maybe a good idea, but its quite a large departure from the current paper.
> > At the current status of the paper, the comment stands.
> >
> > " In practice, when the hypothesis class is misspecified with respect to the ground truth (i.e. sinusoidal ground truth boundary, linear surfaces for the hypothesis class), while we found that while D-PDDM is still effective in monitoring deteriorating changes, the algorithm becomes susceptible to high FPRs when the underlying shift is non-deteriorating."
> > This is actually a substantial weakness of the paper. In practice, we usually do not want to assume that the true model can exactly be accessible to the optimizer.
> >
> > "The probability term here is not induced by the PAC theory under finite samples. ..."
> > I admit that I still do not understand what is the probability space.
> > Are PDD and D-PDD fixed events or probabilistic events?
> > They seem to be fixed (meaning either true or not true) for a given g, f and epsilon_f,
> > because they are defined as statements comparing the (expected) probability of agreement to a bound.
> >
> > Also, in your response, do you mean the event that f=g on (all) examples in a sample of size n from P ?
> >
> > Or is it that there is a probability distribution on f and you discuss the probability that a random f would meet PDD and D-PDD?

---

> ### Author Response · Authors · 2024-11-22
> **Response to Reviewer CJWB (2/2)**
>
> >It is not clear if it is useful if we don't have a very accurate model.
>
> We understand your comment to be referring to the accuracy of the base model on the provided training data. In the context of monitoring a model's deployment in real-world scenarios, it is reasonable to assume that the baseline model performs well on the training data. This assumption aligns with standard deployment practices, as models that perform poorly during training (e.g., achieving only 50%-60% accuracy on binary classification tasks, for instance) are typically not deployed. Instead, such cases would prompt further investigation into the training process.
>
> Although theoretically, there is a risk in running Algorithm 1 and Algorithm 2 using a poorly trained base classifier as discussed in 4.3.1., in further experiments using the Bayesian implementation, we surprisingly found that the D-PDDM test is still able to successfully identify deteriorating shifts when they occur. Discussion around this finding will be provided in the revised version of the supplementary section.
>
> >Isn’t it too good to be to be true to assume that g the ground truth function is in H? At first seemed to me a technicality, but when I look at the proof fo Lemma 2.1 (in the supplementary), PDD->DPDD, it hedges on g being the that function. But is g achievable in practice? and if not, what are implications?
>
> We agree that the proof in g can be replaced by the invariant conditional distribution if we assume that the predictor is stochastic. The proof included here is designed for the sake of simplicity but it can be extended to the general probabilistic setting. In practice, when the hypothesis class is misspecified with respect to the ground truth (i.e. sinusoidal ground truth boundary, linear surfaces for the hypothesis class), while we found that while D-PDDM is still effective in monitoring deteriorating changes, the algorithm becomes susceptible to high FPRs when the underlying shift is non-deteriorating. These results will further be mentioned in the supplementary section on Bayesian D-PDDM to be included as discussed previously.
>
> >I don't quite understand what is the probabilistic statement hiding here "With high probability, PDD is equivalent to D-PDD". This is I think a similar probability model as here "As this equivalence happens in probability, Def. 2 allows for certain false positive errors w.r.t. to Def. 1" and here "We need to demonstrate h = g with high probability" (in proof of the Equivalence condition, Lemma A.1). But to me It seems that expressions such as "err(f, Qg)" and "err(f,Pg)" are not supposed to be probabilistic statements, because f and g are fixed functions and this is an expected statements.”
>
> The probability term here is not induced by the PAC theory under finite samples. We note here that since $err(f, P_g) < \epsilon$ and that the loss function is binary, we have that under probability $P$, if $P(f \neq g) <\epsilon$, then the event $f = g$ in distribution $P$ will be satisfied with high probability.
>
> ### Closing comments
>
> We thank the reviewer again for their detailed and valuable feedback on our work. We hope that our rebuttal addresses their questions and concerns. A revised copy of our submission and supplementary will be uploaded and notified via the global response thread in the coming days. We kindly ask the reviewer to consider a fresher evaluation of our paper if the reviewer is satisfied with our responses. We are also more than happy to answer any further questions that arise that might improve your assessment of the work.

---

### Official Review · Reviewer_jbNA · 2024-11-07

**Soundness:** 2
**Presentation:** 2
**Contribution:** 2
**Rating:** 3
**Confidence:** 4

**Summary:**

This paper proposes D-PDDM, a new unsupervised algorithm designed to detect post-deployment deterioration in machine learning models. D-PDDM monitors prediction disagreements between the deployed model and auxiliary models using test data, without requiring labeled testing data or training data. Experimental results on both synthetic and real clinical datasets demonstrate that the proposed method outperforms existing baselines.

**Strengths:**

1. The paper tackles the critical problem of monitoring distribution shifts after deployment, which is essential for the reliable application of AI models in real-world scenarios.

2. The authors provide theoretical backing for the proposed method, contributing to a more rigorous understanding of its behavior.

**Weaknesses:**

1. The theoretical analysis has significant limitations: It is restricted to noiseless binary classification settings where the ground-truth labeling function is deterministic. This assumption is overly stringent and does not apply to many real-world classification problems. Moreover, according to Lemma 2.1, the method is only applicable under the assumption of covariate shift, which is a restrictive condition that does not generally hold in practice.

2. The literature review is not comprehensive. The authors overlook important work in source-free and test-time adaptation, areas that feature methods for detecting and adapting to distribution shifts, such as entropy minimization and uncertainty quantification. The lack of comparison with these methods significantly limits the technical contribution of the paper.

3. The discussion of auxiliary models in the problem setting section lacks sufficient detail. There is no clear explanation of the basic requirements or conditions for selecting or configuring auxiliary models. This leaves readers unclear about how these models are chosen or how they relate to the proposed method. Additionally, a high-level overview of the unsupervised baselines used for comparison would help provide better context for understanding the unique contributions of D-PDDM.

4. The experimental section does not address the robustness of D-PDDM with respect to the number and efficiency of auxiliary classifiers. While D-PDDM does not require training data during deployment, it does introduce additional auxiliary classifiers. However, the authors do not discuss the extra storage and inference costs associated with these models. This oversight undermines the claimed advantage of the method, as the practical impact of these overheads on deployment efficiency remains unclear.

5. The experimental design lacks critical information. There are no data statistics provided for the synthetic and real-world clinical datasets. The hyperparameter tuning and model selection processes are not described, raising concerns about the fairness and generalizability of the benchmark.

6. No code or data is provided for reproducibility. Although the authors include a link to their code repository, it leads to an empty repository, making it impossible to replicate the results presented in the paper.

**Questions:**

Please see Weaknesses section.

---

> ### Author Response · Authors · 2024-11-22
> **Response to Reviewer jbNA (1/3)**
>
> We would like to thank the reviewer for their helpful comments. We greatly appreciate their time and effort, and are encouraged that they find that our method tackles a problem “essential for the reliable deployment of AI models”, and that "theoretical backing for the proposed method" was provided. In the following we hope to address some concerns raised in their review.
>
> >The theoretical analysis has significant limitations: It is restricted to noiseless binary classification settings where the ground-truth labeling function is deterministic. This assumption is overly stringent and does not apply to many real-world classification problems. Moreover, according to Lemma 2.1, the method is only applicable under the assumption of covariate shift, which is a restrictive condition that does not generally hold in practice.
>
> Thank you for raising this point. We respectfully disagree that the theoretical assumptions in our work significantly limit its contribution. Theoretical analysis in machine learning often requires simplifying assumptions to derive meaningful insights, as demonstrated in many impactful works. For instance, foundational papers like [1, 2, 3] analyze settings based on deterministic ground-truth functions, much like our own assumptions. Moreover, the assumption of covariate shift remains a widely studied and practically relevant setting. Many recent theoretical advancements, including those in [1, 2], similarly focus on covariate shifts to establish rigorous guarantees and provide actionable insights.
>
> While we do agree with the reviewer that many real-world scenarios are the result of a complex interplay between their causal factors, giving rise to various mixtures of shifts, we remark that shifts that preserve their underlying causal dynamics such as those due to differences in measuring instruments, social phenomena, behavioral changes, and seasonal trends, have been effectively modeled under the framework of deteriorating shift.  In addition, many empirical works following the benchmarks proposed in [5] and [6] extend the frontiers of deep models in achieving high performance when confronted with covariate shifts. Thus, the existence of original work with stringent assumptions does not preclude the creation of practical solutions but rather provides a necessary foundation for future research to relax these assumptions and address more complex scenarios.
>
> We also emphasize the gap between theory and practice in machine learning is inherently challenging and not unique to our work. Even in cutting-edge domains such as transformers and large language models (LLMs), recent theoretical studies often rely on basic, idealized assumptions (e.g., [4] analyzes a single-layer transformer model). These simplifications are necessary for advancing theoretical understanding, even if they do not fully encompass real-world complexities.
>
> ### References
>
> [1]  Acuna, David, et al. "f-domain adversarial learning: Theory and algorithms." International Conference on Machine Learning. PMLR, 2021.
>
> [2]  Rosenfeld, Elan, and Saurabh Garg. "(Almost) Provable Error Bounds Under Distribution Shift via Disagreement Discrepancy." Advances in Neural Information Processing Systems 36 (2023): 28761-28784.
>
> [3] Zhao, Shengjia, et al. "Comparing distributions by measuring differences that affect decision making." International Conference on Learning Representations. 2022. (Outstanding Paper)
>
> [4] Bietti, Alberto, et al. "Birth of a transformer: A memory viewpoint." Advances in Neural Information Processing Systems 36 (2024). (Spotlight)
>
> [5] Recht, Benjamin, et al. "Do ImageNet classifiers generalize to ImageNet?." International conference on machine learning. PMLR, 2019. (Oral)
>
> [6] Koh, Pang Wei, et al. "Wilds: A benchmark of in-the-wild distribution shifts." International conference on machine learning. PMLR, 2021.

---

> > ### Comment · Reviewer_jbNA · 2024-11-26
> > **Response to authors' rebuttal**
> >
> > Thanks the authors for providing the rebuttal. After reviewing it, I still have several concerns as follows.
> >
> > - Covariate shift is a well-studied topic with a rich body of theoretical literature. Therefore, in my view, the theoretical contributions in this paper do not offer significant new insights compared to existing works.
> >
> > - I initially requested a comparison with source-free and test-time adaptation methods because these approaches can technically be applied to the proposed setting. The authors claim that model performance can degrade if the model is updated with new data in unnecessary scenarios. Therefore, comparison with source-free and test-time adaptation methods is crucial to verify this claim and strengthen the proposed contributions.
> >
> > - I asked for additional information to clarify the method and experimental design, but this was not provided in the discussion phase. The authors mentioned that these details would be added in the revised version, but they were not addressed in the rebuttal.
> >
> > - A submission should be complete by the paper submission deadline, which includes all supplementary materials and source code. Therefore, I cannot evaluate any code uploaded after the submission deadline, and my assessment of the reproducibility of this submission remains based on the materials provided at the time.
> >
> > In summary, the rebuttal does not fully address my concerns, so I have decided to maintain my original score.

---

> ### Author Response · Authors · 2024-11-22
> **Response to Reviewer jbNA (2/3)**
>
> >The literature review is not comprehensive. The authors overlook important work in source-free and test-time adaptation, areas that feature methods for detecting and adapting to distribution shifts, such as entropy minimization and uncertainty quantification. The lack of comparison with these methods significantly limits the technical contribution of the paper.
>
> Thank you for bringing up the domain of source-free and test-time adaptation. While related, we would like to highlight the key distinctions between these adaptation methods and our approach to post-deployment monitoring.
> - **Different Objectives**. The primary goal of source-free and test-time adaptation is to design methods for adapting a model to an unlabeled target domain. These methods generally assume that a performance drop has already occurred due to distribution shift, necessitating adaptation. In contrast, our paper addresses a guardrail problem preceding adaptation: to determine whether the target distribution actually leads to performance degradation. If no degradation is detected, adaptation is unnecessary. If degradation is detected, the system can stop operation and query humans for appropriate intervention such as initiating adaptation or retraining.
>
> - **Deployment Data-Dependent Monitoring**. Different from traditional out-of-distribution (OOD) detection methods which focus solely on identifying distribution shifts without regard to their impact on performance, our monitoring algorithm is adapted to assess whether these shifts cause performance deterioration. To ensure completeness, as per the reviewer’s suggestion, we are currently running comparisons with traditional OOD detection methods and plan to add to our rebuttal discussions prior to its end. These methods, however, tend to generate significant false alarms in scenarios involving non-deteriorative shifts, as demonstrated in Result 1. In contrast, our monitoring algorithm is specifically designed to handle such shifts robustly, avoiding unnecessary alerts and enabling more reliable post-deployment operation.
>
> Furthermore, to reinforce the completeness of our method evaluation as suggested by the reviewer, we will include two additional baselines for the synthetic comparisons: the Relative Mahalanobis distance [1] and the ODIN baseline [2]. The results will be published upon completion in the global rebuttal as well as in this rebuttal thread in the upcoming days.
>
> >The discussion of auxiliary models in the problem setting section lacks sufficient detail. There is no clear explanation of the basic requirements or conditions for selecting or configuring auxiliary models. This leaves readers unclear about how these models are chosen or how they relate to the proposed method. Additionally, a high-level overview of the unsupervised baselines used for comparison would help provide better context for understanding the unique contributions of D-PDDM.
>
> We appreciate the reviewer for raising this concern about the selection of the auxiliary models. The omitted details in the main paper regarding their configuration will be added in the experiments section of the supplementary section, and we will ensure proper reference from Section 3 to it. Specifically, in the supplementary:
> - We will append details regarding the Bayesian perspective where we view the restricted hypothesis space Hp as a posterior distribution, effectively casting the pretraining algorithm (Algorithm 1) into training Bayesian neural networks and sampling from its posterior distribution.
> - Details and discussions regarding approximate sampling schemes as well as suggested best practices will be included.
>
> Regarding details on the unsupervised baselines, we will make the reference to their respective sections in the supplementary clear. Furthermore, as the reviewer suggested, we will provide discussion on how D-PDDM leverages information given by the base model in order to resist flagging false positives in the non-deteriorating shift scenario while unsupervised baselines indiscriminately flag shifts, thereby providing context for the unique contributions of D-PDDM.
>
> ### References
>
> [1] Ren, Jie, et al. "A simple fix to mahalanobis distance for improving near-ood detection." arXiv preprint arXiv:2106.09022 (2021).
>
> [2] Liang, Shiyu, Yixuan Li, and Rayadurgam Srikant. "Enhancing the reliability of out-of-distribution image detection in neural networks." arXiv preprint arXiv:1706.02690 (2017). ICLR 2018

---

> ### Author Response · Authors · 2024-11-22
> **Response to Reviewer jbNA (3/3)**
>
> > The experimental design lacks critical information. There are no data statistics provided for the synthetic and real-world clinical datasets. The hyperparameter tuning and model selection processes are not described, raising concerns about the fairness and generalizability of the benchmark.
>
> Thank you for pointing these out! The description of the synthetic data generation process is located in Appendix B.1. including the statistics used for the experiments. Regarding details on the GEMINI clinical dataset, we will further improve section B.5 with more details on the splits selection for each temporal range for the time-varying experiments, as well as the patient data splitting and mixing process for the age-shift experiments.
>
> > No code or data is provided for reproducibility. Although the authors include a link to their code repository, it leads to an empty repository, making it impossible to replicate the results presented in the paper.
>
> Thank you for pointing this out! The code is uploaded to the appropriate repository. The hyperlink is included here for convenience: https://anonymous.4open.science/r/d_pddm-F966/
>
> ### Closing comments
>
> We reiterate once more here that results on new baselines will be published upon completion in the global rebuttal as well as in this rebuttal thread in the upcoming days. Furthermore, a revised copy of our submission and supplementary will be uploaded and notified via the global thread.
>
> We thank the reviewer again for their valuable and detailed feedback on our work. We hope that our rebuttal addresses their questions and concerns. We kindly ask the reviewer to consider a fresher evaluation of our paper if the reviewer is satisfied with our responses. We are also more than happy to answer any further questions that arise that might improve your assessment of the work.

---

### Meta-Review · Area_Chair_Fm7b · 2024-12-15

**Metareview:**

In this work, the authors propose D-PPDM to monitor the performance of machine learning models post-deployment. The method uses auxiliary models to detect when shifts in data distribution lead to performance deterioration, while being robust to shifts that do not affect model performance negatively.

The reviewers recognized the importance of this topic and found strengths in the proposed strategy (no need for labelled test data or training data access). However, they also pointed weaknesses of the method, namely the fact that one or more auxiliary models need to be accessible. This has computing and storage costs that have not been satisfactorily addressed by the authors. In addition, there was little discussion on the effect of model under- or over-fitting in these auxiliary models, and the reviewers pointed to missing baselines. For these reasons, I believe the paper needs more work and recommend rejection.

**Additional Comments On Reviewer Discussion:**

The authors provided responses to the reviewers' concerns, but these lacked clear changes that would be made to the text and did not convince the reviewers to modify their scores. Beyond the initial rebuttal, the authors did not further engage.

---

### Decision · Program_Chairs · 2025-01-22

Reject